# Targeting myeloid derived suppressor cells reverts immune suppression and sensitizes BRAF-mutant papillary thyroid cancer to MAPK inhibitors

Peitao Zhang[1,2,7], Haixia Guan [3,4,7], Shukai Yuan[1,7], Huili Cheng[1], Jian Zheng[5], Zhenlei Zhang[1], Yifan Liu[1], Yang Yu[6], Zhaowei Meng[2], Xiangqian Zheng[6] & Li Zhao [1✉]

MAPK signaling inhibitor (MAPKi) therapies show limited efficacy for advanced thyroid cancers despite constitutive activation of the signaling correlates with disease recurrence and persistence. Understanding how BRAF pathway stimulates tumorigenesis could lead to new therapeutic targets. Here, through genetic and pathological approaches, we demonstrate that BRAF$^{V600E}$ promotes thyroid cancer development by increasing myeloid-derived suppressor cells (MDSCs) penetrance. This BRAF$^{V600E}$-induced immune suppression involves re-activation of the developmental factor TBX3, which in turn up-regulates CXCR2 ligands in a TLR2-NFκB dependent manner, leading to MDSCs recruitment into the tumor microenvironment. CXCR2 inhibition or MDSCs repression improves MAPKi therapy effect. Clinically, high TBX3 expression correlates with BRAF$^{V600E}$ mutation and increased CXCR2 ligands, along with abundant MDSCs infiltration. Thus, our study uncovers a BRAF$^{V600E}$-TBX3-CXCLs-MDSCs axis that guides patient stratification and could be targeted to improve the efficacy of MAPKi therapy in advanced thyroid cancer patients.

---

[1] The Province and Ministry Co-sponsored Collaborative Innovation Center for Medical Epigenetics, Key Laboratory of Immune Microenvironment and Disease (Ministry of Education), Department of Biochemistry and Molecular Biology, School of Basic Medical Sciences, Tianjin Medical University, Tianjin, China. [2] Department of Nuclear Medicine, Tianjin Medical University General Hospital, Tianjin, China. [3] Department of Endocrinology, Guangdong Provincial People's Hospital, Guangdong Academy of Medical Sciences, Guangzhou, Guangdong Province, China. [4] The Second School of Clinical Medicine, Southern Medical University, Guangzhou, Guangdong Province, China. [5] Department of Immunology, School of Basic Medical Sciences, Tianjin Medical University, Tianjin, China. [6] Department of Thyroid and Neck Oncology, National Clinical Research Center for Cancer; Key Laboratory of Cancer Prevention and Therapy, Tianjin's Clinical Research Center for Cancer, Tianjin Medical University Cancer Institute and Hospital, Tianjin, China. [7] These authors contributed equally: Peitao Zhang, Haixia Guan, Shukai Yuan. ✉email: shzhaoli@tmu.edu.cn

As the most common type of differentiated thyroid cancer (DTC), the incidence of papillary thyroid cancer (PTC) has increased since the early 1980s and become the fastest-growing cancer in most countries. Despite the overall favorable prognosis of this disease, 20–30% of patients experience recurrence, even 5–10% have advanced disease[1–3]. BRAF$^{V600E}$, the most common genetic mutation associated with 40–80% PTC, activates MAPK signaling and impairs thyroid lineage factor expression, thus defines less differentiated cell state and correlates with cancer recurrence and persistence[4,5]. Nevertheless, targeted therapies with MAPK inhibitor (MAPKi), including Vemurafenib or Dabrafenib, exhibited reserved efficacy on BRAF$^{V600E}$-positive metastatic, RAI-refractory DTC patients due to primary or acquired drug resistance[6]. As such, further understanding the molecular mechanisms of tumor progression and developing effective adjuvant therapies are of the utmost importance for improving disease-free survival (DFS) in advanced PTC patients.

Nowadays, the cancer immune therapy field has been revolutionized with growing understanding of the dynamic interaction between immune system and tumor microenvironment (TME). In PTC patients, BRAF$^{V600E}$ mutation correlates with suppressive immune microenvironment represented by high levels of immune checkpoint regulators Programmed death ligand 1 (PD-L1), Cytotoxic T-lymphocyte antigen 4 (CTLA-4), as well as Indoleamine 2,3-dioxygenase (IDO)[7–9]. In contrast, tumor-specific major histocompatibility complex class I and II (tsMHCI and tsMHCII) required for immune cell recruitment are significantly repressed in BRAF$^{V600E}$ PTC cells, correlating with reduced lymphocytic infiltration of CD8$^+$ and CD4$^+$ T cells[9–11]. Our previous study found that combined therapy of BRAF$^{V600E}$ inhibitor PLX4032 and anti-PD-L1 antibody inhibited PTC development more efficiently than either single treatment, partially through restoring tsMHCII level[12]. Clinical trials combining MAPKi with immune therapy, or aiming at depleting and repolarizing tumor-associated macrophages (TAM) to enhance anti-tumor immune response for advanced thyroid tumors are also currently on going[13].

As a heterogeneous population of immature myeloid cells, myeloid-derived suppressor cells (MDSCs) function as the major orchestrator of immune-suppressive environment in cancer. MDSCs are recruited into the TME mainly by CXCR2 ligands, which are significantly increased in various cancers[14,15]. Consequently, blockage of CXCR2 by genetic ablation or pharmacologic inhibition reduces tumorigenesis and metastasis, suggesting that MDSCs could be a promising therapy target[16–18]. In PTC, circulating MDSCs were reported to be elevated[19,20]. Studies also showed that PTC and anaplastic thyroid cancer (ATC) cells secrete CXCL8, a representative CXCR2 ligand chemokine, which promotes tumor cell stemness and EMT, thus correlates with thyroid cancer advancing[21]. Besides these evidences, whether MDSCs and other types of immune cells are recruited into thyroid cancer microenvironment, and, if so, what pathological roles they are playing, or how activated BRAF/MAPK pathway regulates this process, remains to be explored.

T-box transcription factor 3 (TBX3), first characterized in ulnar-mammary syndrome, belongs to T-box transcription factor family and plays critical roles in embryonic development and tumorigenesis. TBX3 is generally regarded as an oncogene promoting tumor cell proliferation and metastasis[22,23]. Whether TBX3 has any regulatory function on inflammation-related pathways and TME modulation has not been explored. PTC patient specimen analysis shows that TBX3 is highly expressed in transformed thyrocytes[24]. While early in embryogenesis, TBX3 seems to play an instrumental role during fate determination of thyroid primordium[25]. Besides, TBX3 was up-regulated by BRAF$^{V600E}$-induced MAPK pathway activation and promotes

melanoma migration via repressing E-cadherin, which correlates TBX3 with BRAF$^{V600E}$ associated tumorigenesis[26–28]. Based on these evidences, whether TBX3 participates in BRAF$^{V600E}$-induced thyroid tumorigenesis will not only provide us a better understanding of this specific factor, but also clarify the underlying correlation between organ development and tumorigenesis.

In this work, we investigate the landscape of immune cell distribution in BRAF$^{V600E}$-induced advanced PTCs and find that MDSCs abundance may be a critical determinant for cancer progression. By crossing and analyzing different genetic models, we confirm that transcriptional re-activation of TBX3 is an indispensable molecular event for cancer initiation and progression, whereby it links BRAF/MAPK pathway activation to CXCR2 ligands elicited MDSCs infiltration. We then assess the therapeutic effects of combined MDSCs antagonism and BRAF/MAPK repression. The adjuvant treatments promote recognition and elimination of PTC cells by the immune system in a preclinical mouse model, and could therefore offer an effective therapeutic strategy for advanced PTC patients.

## Results

**Loss of Tbx3 inhibits Braf$^{V600E}$-induced PTC initiation and progression.** BRAF$^{V600E}$-caused constitutive activation of MAPK pathway correlates strongly with PTC advancing and relapse. To search for responsive factors and potential targetable molecular events, we explored several online data resources based on melanoma studies because the oncogenic role of BRAF$^{V600E}$ is well understood and a large body of the mechanism and translational investigations have been conducted. We integrated three published datasets (GSE161430, GSE75299, GSE152699) regarding differentially expressed genes (DEGs) upon MAPKi therapy across distinct species[29,30], and identified DEGs that showed the most significant response to BRAF/MAPK activity. Among the overlapped DEGs, we found TBX3, a transcription factor involved in PTC cell proliferation, was highly responsive to MAPKi (Supplementary Fig. 1a)[24]. As a developmental factor, Tbx3 shows low expression level in adult mouse thyroids. However, whether TBX3 expression pattern changes during BRAF$^{V600E}$-induced transformation is still unknown.

To evaluate TBX3 expression in PTC progression, we generated PTC model by crossing thyroid peroxidase *(TPO)-Cre* with *LSL-Braf$^{V600E}$* (named as mPTC) with *Tbx3$^{GFP/+}$* mice to monitor dynamic expression change of Tbx3[31,32]. The endogenous BRAF$^{V600E}$ expression led to classic PTC formation with deregulated thyroid lineage gene expression and thyroid hormonogenesis by 5 weeks (Supplementary Fig. 1b, c). Strong GFP signal was detected in mPTC tissue in comparison to the weak signal from normal thyroid (Fig. 1a). Consistently, Tbx3 mRNA and protein levels were highly up-regulated within mPTC tumors (Fig. 1b, c and Supplementary Fig. 1d). The high expression of Tbx3 in mPTC tissue suggests up-regulation of Tbx3 may be a pre-requisite for BRAF$^{V600E}$-associated PTC under different pathological stages.

Then, to clarify the biological significance of Tbx3 in PTC development, we deleted *Tbx3* gene by crossing *Tbx3$^{flox/flox}$* mice with mPTC line (homozygotes of the *Tbx3* deletion will be referred to as mPTC/Tbx3$^{-/-}$ in the following text, while, heterozygotes as mPTC/Tbx3$^{+/-}$) (Supplementary Fig. 1e). Strikingly, although tumors were formed locally in mPTC mice thyroids around 5 weeks with 95% or higher penetrance, they are rarely detected in mPTC/Tbx3$^{-/-}$ mice, while mPTC/Tbx3$^{+/-}$ mice developed tumors with middle sizes. Quantification further confirmed that Tbx3 loss inhibited initiation and development of BRAF$^{V600E}$-induced PTC in a dose-dependent manner (Fig. 1d). Histologically, mPTC tumors displayed disrupted follicle

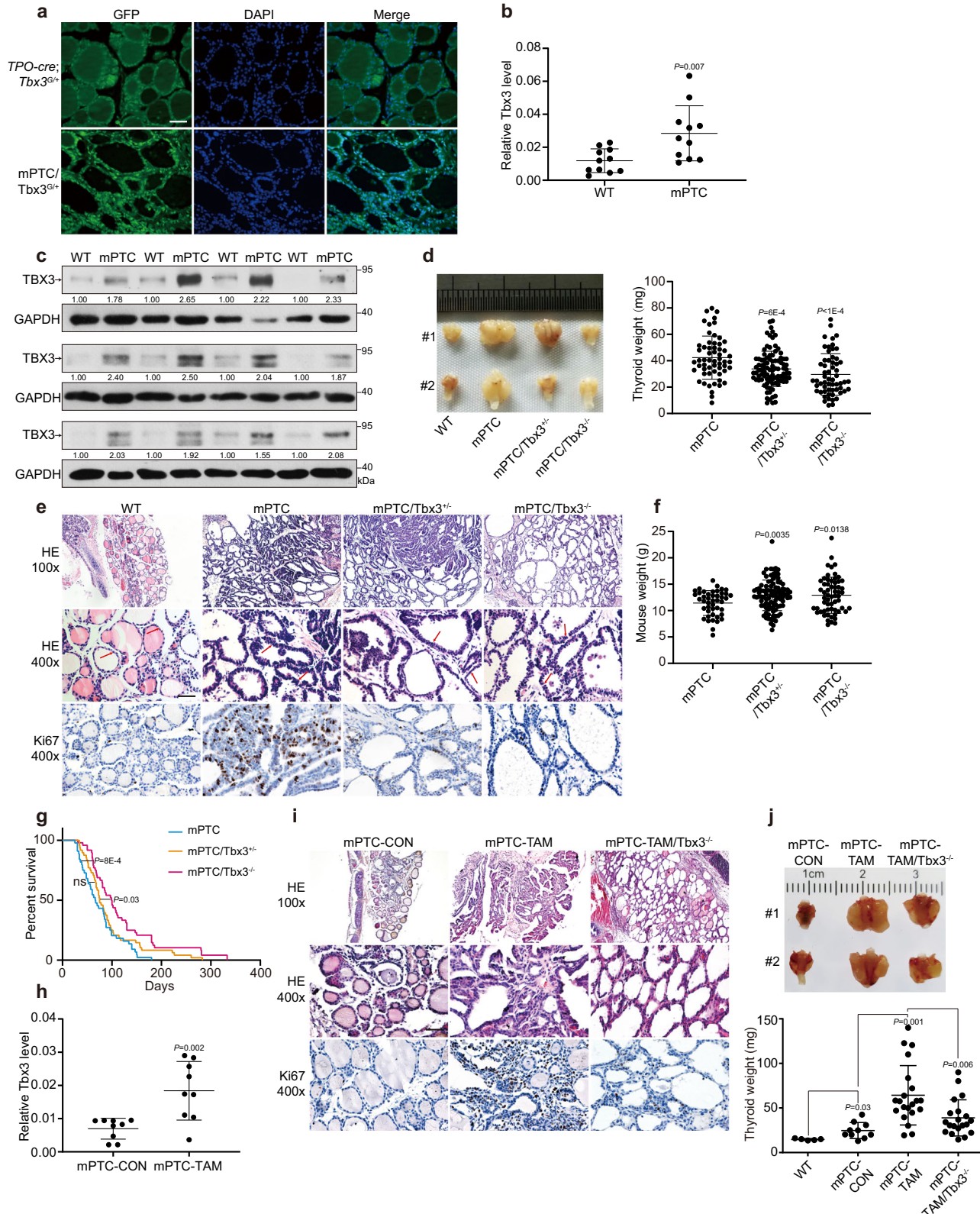

architecture while lined by tall cells with enlarged nuclei (Fig. 1e). In contrast, tumors from mPTC/Tbx3$^{+/−}$ and mPTC/Tbx3$^{−/−}$ littermates, if formed, showed less advanced hyperplasia and retained relatively higher ratio of follicles, with significantly repressed cell proliferation as well in mutant tumors (Fig. 1e and

Supplementary Fig. 1f). Interestingly, Tbx3 reduction relieved the mouse size and weight loss accompanied with mPTC occurance, which may be partially due to restored thyroid lineage function (Fig. 1f and Supplementary Fig. 1g). Consequently, mPTC/Tbx3$^{−/−}$ mice had significantly prolonged survivals compared

**Fig. 1 Tbx3 is necessary for Braf^V600E-induced PTC formation. a** Representative images of TBX3-GFP positive cancer cells analyzed by GFP IF staining in mPTC/Tbx3^G/+ compared with *TPO-cre; Tbx3^G/+* littermates at 5w. Scale bars, 50μm. **b** The expression of Tbx3 was measured by RT-qPCR in thyroid glands from wild-type (WT) and mPTC littermates at 5w, GAPDH was used as the loading control, *n* = 11 pairs of littermates. **c** The expression of TBX3 was measured by western blot with densitometric analyses in thyroid glands from WT and mPTC littermates at 5w, *n* = 12 pairs of littermates. **d** The image representing whole thyroid tissues from 2 pairs of WT, mPTC, mPTC/Tbx3^+/− and mPTC/Tbx3^−/− mice at 5w, and related thyroid weight was plotted, *n* = 58 mPTC, *n* = 95 mPTC/Tbx3^+/−, *n* = 59 mPTC/Tbx3^−/−. **e** Representative H&E and Ki67 IHC staining on thyroid tissues from WT, mPTC, and mPTC/Tbx3^−/− littermates, red arrows points to papillary formations. Scale bars, 50μm. **f** Mouse weight of mPTC, mPTC/Tbx3^+/−, and mPTC/Tbx3^−/− mice at 5w, *n* = 45 mPTC, *n* = 97 mPTC/Tbx3^+/−, *n* = 60 mPTC/Tbx3^−/−. **g** Survival of mice bearing mPTC, mPTC-Tbx3^+/− or mPTC-Tbx3^−/−, *n* = 44 mPTC, *n* = 48 mPTC/Tbx3^+/−, *n* = 48 mPTC/Tbx3^−/−. **h** RT-qPCR analysis of Tbx3 in thyroid tissues from *TPO-creER; Braf^V600E*CA with intraperitoneal tamoxifen injection at 1 m (mPTC-TAM) or oil treatment as control (mPTC-CON), each pair of tissue was obtained at same time, *n* = 9. **i** Representative H&E and Ki67 IHC staining on thyroid tissues from mPTC-CON, mPTC-TAM or mPTC-TAM/Tbx3^−/−. Scale bars, 50μm. **j** Representative images of whole thyroid tissues from mPTC-CON, mPTC-TAM or mPTC-TAM/Tbx3^−/− at 8 m after intraperitoneal tamoxifen injection, *n* = 5 WT, *n* = 10 mPTC-CON, *n* = 21 mPTC-TAM, *n* = 20 mPTC-TAM/Tbx3^−/−. Thyroid tissue weights were analyzed. *n* = 3 biological independent samples (**a, e, i**). Data are shown as the mean ± s.d. (**b, d, f, h, j**). *P* values were calculated by unpaired two-tailed Student's *t* test (**b, d, f, h, j**) or Logrank (Mantel-Cox test) (**g**). Uncropped immunoblots and statistical source data are provided in Source Data.

with mPTC mice (Fig. 1g). Tumor volumes in the lineage tracing line mPTC/Tbx3^G/+ were also slightly reduced, probably due to compromised *Tbx3* dose[32] (Supplementary Fig. 1h). These results show that Tbx3 determines BRAF^V600E-induced thyroid cancer initiation and progression in a dose-dependent way.

Next, we employed the inducible mouse model generated by crossing *TPO-creER* with *LSL-BRAF^V600E* [33]. After tamoxifen induction for 1 month, histological transformation began with follicle enlargement and progressed gradually till aggressive features of PTC became apparent around 8 months (Supplementary Fig. 1i, j). The abundance of Tbx3 was also continuously up-regulated with tumor progression, supporting its important role in the tumor development (Fig. 1h and Supplementary Fig. 1k). Postnatal knock-out of *Tbx3* (referred to mPTC-TAM and mPTC-TAM/Tbx3^−/−) gave rise to the similar phenotype and impeded the malignancy with reduced Ki67 expression (Fig. 1i and Supplementary Fig. 1l). Tumor volume and weight were significantly reduced in mPTC-TAM/Tbx3^−/− as well (Fig. 1j). No significant relief of mouse weight or lineage factor expression was observed probably due to relatively late removal of Tbx3 (Supplementary Fig. 1m). Taken together, these data suggest that Tbx3 is up-regulated and possibly oncogenic during human and murine PTC tumorigenesis.

**BRAF/MAPK directs TBX3 expression through AP-1-mediated transcriptional regulation.** We next aimed to find the mechanism leading to TBX3 up-regulation during BRAF^V600E-induced PTC progression. Similar as observed in mouse models, over-expression of wild-type BRAF or BRAF^V600E in normal thyroid and PTC cells (K1 [BRAF^V600E] and TPC1 [BRAF^WT]) resulted in up-regulation of TBX3 (Fig. 2a). In contrast, knock-down of BRAF, pharmacological inhibition with PLX4032, or inhibition of downstream ERK1/2 with SCH772984 led to down-regulation of TBX3 in a dose-dependent manner, supporting that TBX3 is under the control of BRAF/MAPK pathway (Fig. 2b, c).

As an important cluster of transcription factors, AP-1 proteins transduce the majority of biological function downstream BRAF/MAPK pathway. Genome-wide expression analysis revealed that BRAF^V600E-induced melanoma is accompanied with highly enriched c-Jun, JunB, and c-Fos, we thus wonder whether similar AP-1 factors function in PTC development[26]. Indeed, AP-1 levels were shifted synergistically with BRAF/MAPK activity in PTC cells, and general repression of AP-1 with SR11302 caused TBX3 reduction in a dose-dependent manner[34], both of which support the potential involvement of AP-1 in this context (Supplementary Fig. 2a–c). Next, we manipulated the expression of representative AP-1 factors within different PTC cells and illustrated that TBX3

levels were tightly controlled by AP-1 (Supplementary Fig. 2d–g). Furthermore, over-expression of AP-1 proteins was able to restore TBX3 down-regulation caused by BRAF/MAPK repression in a cumulative manner (Fig. 2d). On the other hand, deprivation of any of the three AP-1 proteins blocked BRAF/MAPK over-activation-induced TBX3 up-regulation (Fig. 2e). Collectively, these findings establish a regulatory axis between BRAF/MAPK, AP-1 proteins, and TBX3 expression.

To investigate how AP-1 regulates TBX3 expression, we analyzed potential AP-1 binding sites on *TBX3* promoter through the Jasper database (Fig. 2f). Using luciferase activity assay of different truncated promoter constructs and chromatin immunoprecipitation (ChIP) assay with anti-Flag, we realized that the -149bp site may play the predominant role (Fig. 2g, h and Supplementary Fig. 2h). Specific mutation and endogenous ChIP experiments also showed that AP-1 proteins were recruited to -149bp site (Fig. 2i, j). In addition, we conducted in vitro DNA binding affinity assay, which further confirmed the binding of AP-1 to -149 site (Supplementary Fig. 2i). Factually, BRAF/MAPK pathway blockage repressed TBX3 promoter activity (Fig. 2k). Thus, we conclude that BRAF/MAPK induces TBX3 up-regulation through AP-1 mediated transcriptional activation.

During the series of biochemical studies, we noticed that BRAF^V600E affected AP-1 and TBX3 more profoundly than wild-type BRAF. In fact, transient DOX-induced BRAF^V600E expression in the mutation-negative TPC-1 cells was enough to increase p-ERK1/2, AP-1, and TBX3 at both mRNA and protein levels, indicating that TBX3 may transduce the oncogenic function downstream of constitutively activated BRAF/MAPK signaling (Supplementary Fig. 2j). Importantly, further analysis using dataset of Thyroid carcinoma (THCA) from The Cancer Genome Atlas (TCGA) (*n* = 501) revealed that TBX3 expression was positively correlated with BRAF levels, suggesting that their functional coupling is clinically important (Fig. 2l).

**TBX3 promotes PTC development via elevating CXCR2 ligands.** To probe the TBX3-associated molecular events in PTC formation, we compared gene expression profiles between mPTC and mPTC/Tbx3^−/− littermate tumors through RNA sequencing (RNA-seq). KEGG analysis demonstrated that the DEGs were involved in vital biological processes. Surprisingly, the top affected pathways were mainly chemokine secretion-associated signalings (cytokine-receptor interaction or TNF signaling pathway), especially those related to down-regulated genes (Fig. 3a and Supplementary Fig. 3a). Integrated with RNA-seq data from human PTC cells with TBX3 knock-down, inflammation-associated cytokines and chemokines again took up the largest percentage of the jointly altered genes (Fig. 3b, Supplementary

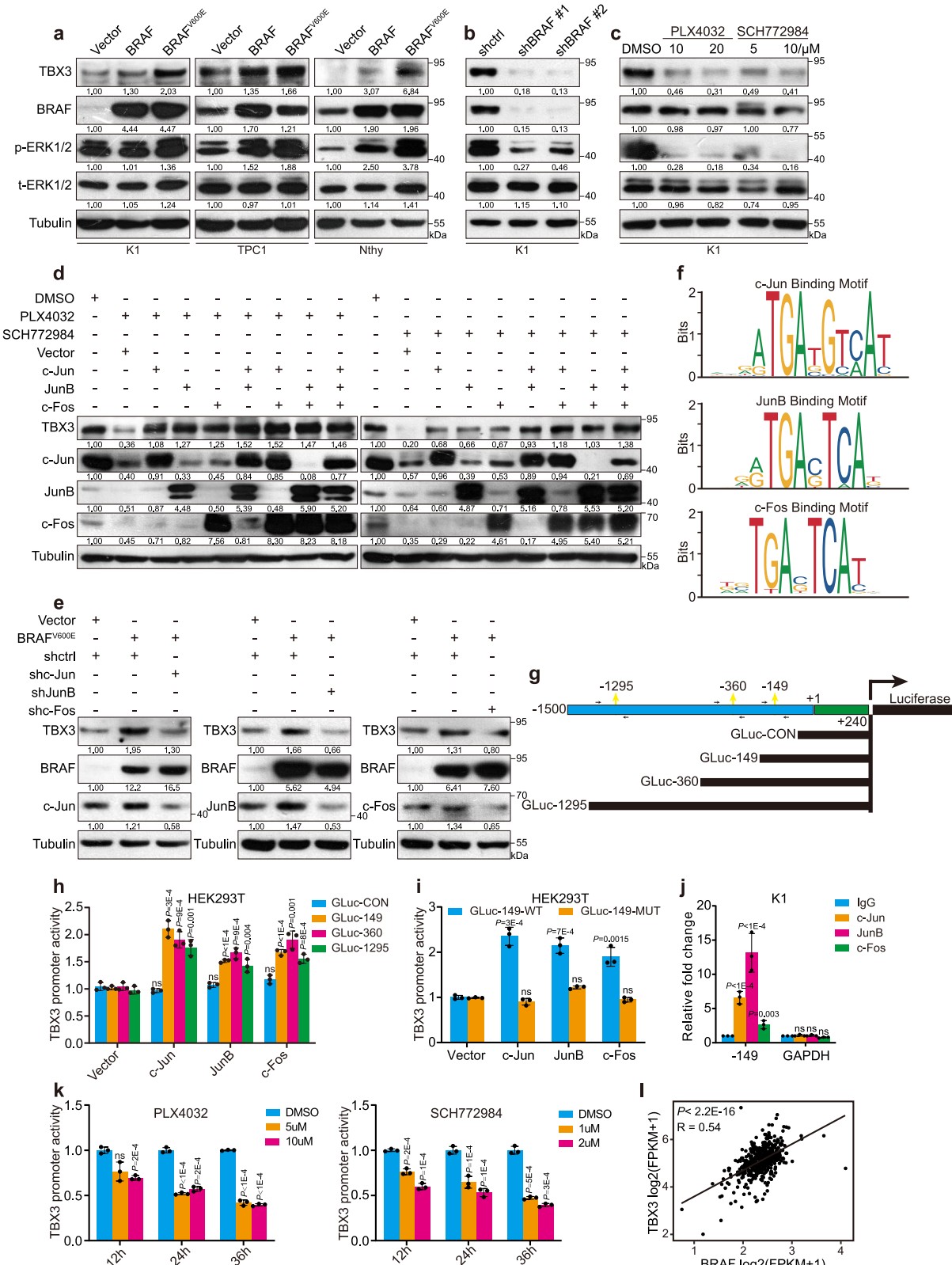

Fig. 3b and Supplementary Data 1). During validation of representative genes by RT-qPCR, we confirmed de-repression of proliferation restricting genes including Cdkn1c, Cdkn2c, and Klf9 upon Tbx3 knock-out, and repression of oncogenic genes such as Igfbp3, Mmp9, Met and Six4 (Fig. 3c). More interestingly, expression of CXCL chemokine members including CXCL1, 2, 3 and CXCL8 as well as a series of inflammatory factors were dramatically decreased in mPTC/Tbx3$^{-/-}$ tumors and TBX3 knock-down PTC cells (Fig. 3d, e). To clarify if the chemokines were down-regulated by Tbx3 knock-out in tumor cells, we crossed *Rosa26-mTmG* reporter line into the mPTC animal and obtained mPTC$^{GFP}$ cells through FACS sorting (Supplementary

**Fig. 2 BRAF/MAPK/AP-1 axis promotes TBX3 transcription in PTC. a, b** Western blot of TBX3 in PTC or normal thyroid cells (Nthy-ori-3-1) with BRAF or BRAF$^{V600E}$ over-expression (**a**) or K1 cells with BRAF knock-down (**b**). **c** K1 cells were treated with BRAF inhibitor PLX4032 or ERK1/2 inhibitor SCH772984 for 24 h, the level of TBX3 was analysis by western blot. **d** TBX3 and AP-1 factors levels in K1 cells over-expressing c-Jun, JunB, or c-Fos, and treated with PLX4032 or SCH772984. **e** TBX3 and AP-1 factors levels in K1 cells with BRAF$^{V600E}$ over-expression and AP-1 knock-down. **f** AP-1 binding motifs were analyzed through TBX3 promoter region using Jasper database. **g** Construction of TBX3 promoter GLuc with potential AP-1-binding sites. The predicted -358bp and -362bp site were overlapped as -360bp site. Arrows represented primers were used for qPCR. **h, i** Promoter GLuc activities were analyzed in HEK293T cells co-transfected with TBX3 truncated promoter (**h**) or TBX3-149-MUT promoter (**i**) and AP-1 expression constructs. **j** Quantitative ChIP (qChIP) analysis with anti-IgG, c-Jun, JunB and c-Fos for -149 site in K1 cells, primers targeting GAPDH gene were used as control. **k** Luciferase activities of GLuc-149 were analyzed in K1 cells treated with PLX4032 or SCH772984 for 12 h, 24 h or 36 h. **l** THCA datasets were downloaded from TCGA database to analyze the correlation between TBX3 and BRAF, $n = 501$. Densitometric analyses of western blot were shown (**a-e**). Two independent experiments were carried out with similar results for each kind of cells (**a-c**), and $n = 3$ biological independent samples (**d, e, h-k**). Data are shown as the mean ± s.d. (**h-k**). $P$ values were calculated by unpaired two-tailed Student's $t$ test (**h-k**). Uncropped immunoblots and statistical source data are provided in Source Data.

Fig. 3c–e). Distinct from the strong Thyroglobulin (Tg) intensity in thyrocytes, primary mPTC$^{GFP}$ cells showed moderate Tg expression, which correlated with the compromised thyroid function and depressed Tg level in mPTC model (Supplementary Fig. 3f). Quantified with RT-qPCR, Cxcls were significantly repressed in primary mPTC/Tbx3$^{-/-GFP}$ cells (Supplementary Fig. 3g).

To obtain a more comprehensive understanding of cytokines influenced by TBX3, cytokine antibody array was applied to serum-deprived media of PTC cells (Fig. 3f, Supplementary Data 2). Compared with the control media, several chemokines were decreased in TBX3 knock-down group. Representative members of CXCR2 ligands, especially CXCL1, 2 and CXCL8, showed dramatic and consistent reduction upon TBX3 removal. Loss of TBX3 in independent cell lines also resulted in significant repression of CXCR2 ligands at mRNA and protein levels (Fig. 3g and Supplementary Fig. 3h, i). In contrast, over-expression of TBX3 significantly up-regulated CXCR2 ligands to various extents (Fig. 3h). Due to the absence of Cxcl8 in mouse, we examined the expression of Cxcl1 and Cxcl2, which were decreased in mPTC/Tbx3$^{+/-}$ and mPTC/Tbx3$^{-/-}$ tumors (Fig. 3i). Thus, TBX3 directs CXCR2 ligands expression in both mouse and human PTC development.

**TBX3 induces CXCR2 ligands expression through the TLR2-NF-κB pathway.** We next asked how TBX3 regulates chemokine expression by blocking various pathways known to mediate chemokine expression[35–38]. Strikingly, inhibition of IKKβ/NF-κB signaling efficiently reduced ectopic TBX3-induced up-regulation of chemokines in thyroid cells. In comparison, repression of p38/CREB, JNK/AP-1, or AKT/STAT3 activity did not show meaningful impact of TBX3-promoted chemokines even though these factors were under TBX3 regulation and have been involved in chemokine regulation (Fig. 4a and Supplementary Fig. 4a–c). Similar regulations were observed on endogenous chemokines in PTC cells (Supplementary Fig. 4c). We further found that deprivation of IKKβ/NF-κB activity with pairs of shRNA targeting IKBKB or RELA exhibited comparable function (Fig. 4b and Supplementary Fig. 4d, e). Indeed, NF-κB pathway members were regulated by TBX3 in PTC cells (Fig. 4c). In vivo, the level of p-IKKα/β, p-p65 were quite sensitive to Tbx3 dose and were eliminated in mPTC/Tbx3$^{-/-}$ tumors (Supplementary Fig. 4f). Taken together, these results suggest that TBX3 induces CXCR2 ligands expression through activating IKKβ/NF-κB pathway.

TBX3 is generally thought to act as a transcription factor to either activate or repress gene expression. As far as the transcriptional repressor role of TBX3 was concerned, we were not able to locate any negative regulators of IKKβ/NF-κB pathway that were directly repressed by TBX3 through RT-qPCR detection (Supplementary Fig. 4g). While, as a transcriptional activator,

TBX3 over-expression elevated CXCR2 ligands in breast cancer as well, indicating similar regulatory mechanisms were involved. We thus compared TBX3-regulated genes in our RNA-seq screening and online breast cancer datasets (Fig. 4d and Supplementary Data 3). Among the co-regulated genes, TLR2, a critical upstream initiator of IKKβ/NF-κB pathway, was found to be positively regulated by TBX3 in both datasets (Fig. 4e and Supplementary Fig. 4h). Functional validation showed that TLR2 knock-down in PTC cells reduced expression of CXCR2 ligands (Supplementary Fig. 4i). Importantly, TLR2 loss impeded TBX3-induced chemokines up-regulation due to interfered p-IKKα/β and p-p65 activation (Fig. 4f, g). Moreover, over-expression of TLR2 restored chemokines down-regulated by TBX3 deficiency (Supplementary Fig. 4j). Luciferase activity assay and ChIP experiments showed that TBX3 may transcriptionally activate TLR2 expression by directly binding to the promoter using the activation domain. (Fig. 4h, i). Therefore, binding of TBX3 to TLR2 was notably repressed in TBX3 deficient cells in comparison to control cells (Fig. 4j). Overall, we determine a positive regulatory axis from TBX3 to CXCR2 ligands mediated by TLR2-NF-κB pathway.

**CXCR2 ligands mediate TBX3-promoted tumor cell proliferation.** The biological function of chemokines, particularly the CXCR2 ligands described above, has barely been addressed in PTC development. Motivated by the restricted cell proliferation in mPTC/Tbx3$^{-/-}$ tumor and reduced CXCR2 ligands in TBX3 deficient cells, we investigated the influence of these chemokines on cell proliferation. The same shRNAs were used to knock down CXCL1 and CXCL2, since they share 90% sequence homology[39]. Decreased expression of CXCL1/2 and CXCL8 significantly inhibited PTC cell proliferation, even cells showed different sensitivity to chemokine doses (Supplementary Fig. 5a and Fig. 5a, b). Remarkably, over-expression of any of the three chemokines restored the proliferation arrest caused by TBX3 loss, suggesting that these chemokines could be regulating cell proliferation downstream of TBX3 (Fig. 5c, d and Supplementary Fig. 5b). Furthermore, we found that CXCR2 inhibitors SCH527123 and Reparixin inhibited PTC cell proliferation in dose-dependent manners even though K1 cells and TPC1 cells responded to different concentrations (Supplementary Fig. 5c, d). Elevated cell proliferation caused by CXCL1, CXCL2, and CXCL8 over-expression was also reduced by CXCR2 inhibitors, indicating that CXCR2 ligands from PTC cells interact with CXCR2 to promote cell proliferation (Supplementary Fig. 5e).

We further validated the above findings using xenograft models. Consistent with genetic model (Fig. 1d, j), tumors were barely formed after TBX3 knock-down (Fig. 5e). However, ectopic expression of any CXCR2 ligand was able to rescue tumor growth and brought back tumor volumes/weights closer to

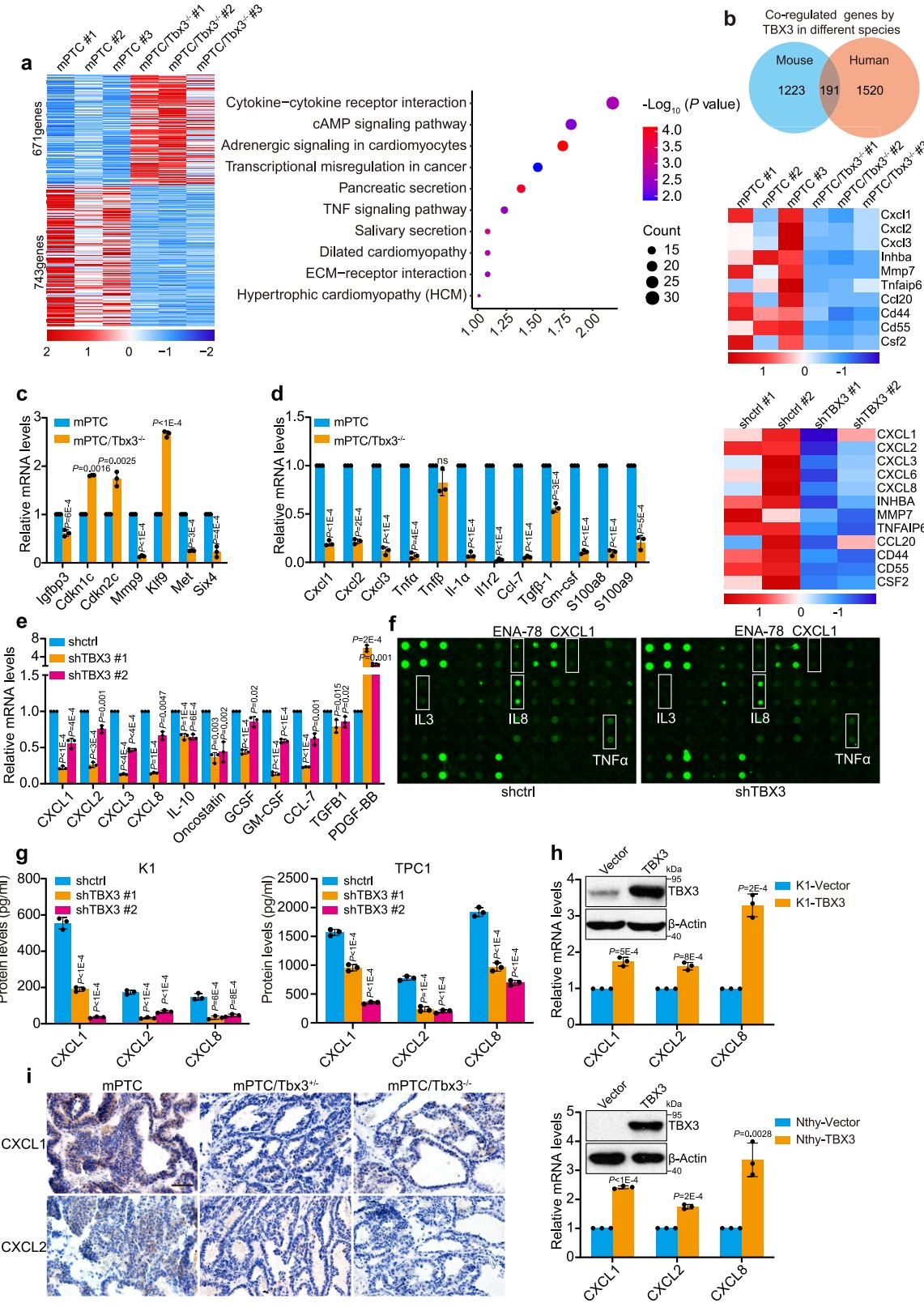

control group (Fig. 5e). Histologically, CXCL1, CXCL2, and CXCL8 were dramatically reduced in TBX3 deficient tumors simultaneously with Ki67 (Fig. 5f). While CXCR2 ligands overexpression led to different extents of Ki67 recovery. Taken together, these results support the hypothesis that TBX3 promotes PTC cell proliferation partially through directing

CXCR2 ligands expression, which in turn function in an autocrine way.

**TBX3 increases MDSCs infiltration and suppresses anti-tumor immunity.** Given the direct regulation of TBX3 on chemokines expression, we reasoned that TBX3 may participate in the

**Fig. 3 Loss of TBX3 leads to CXCR2 ligands reduction. a** Heatmap representation of DEGs (fold change >1.5; P < 0.05) in tumor tissues from mPTC and mPTC/Tbx3$^{-/-}$. The right panel showed the KEGG pathway analysis of DEGs. Data were analyzed using DAVID 6.8 website. **b** Venn diagram analysis across two groups of DEGs from mPTC tissues versus human K1 cells due to TBX3 loss. The heatmap of regulated genes was analyzed according to venn diagram, the upper for mouse and the lower for human. Differentially expressed human genes: 706 up-regulated genes and 1004 down-regulated genes. **c** The expression of genes from mPTC RNA-seq was confirmed by RT-qPCR. **d, e** RT-qPCR confirmation of inflammatory factors among DEGs from mPTC tissue (**d**) or human K1 cells (**e**) RNA-seq. **f** Cytokine antibody array on cell culture media from K1 cells with TBX3 knock-down. **g** ELISA analysis of CXCL1, 2 and CXCL8 in cell culture media from K1 or TPC1 cells with TBX3 knock-down. **h** RT-qPCR analysis of CXCL1, 2 and CXCL8 in K1 and Nthy cells with TBX3 over-expression. **i** IHC staining of CXCL1 and CXCL2 in mPTC, mPTC/Tbx3$^{+/-}$ and mPTC/Tbx3$^{-/-}$. Scale bars, 50μm. $n = 3$ biological independent samples (**c–e, g–i**). Data are shown as the mean ± s.d. (**c–e, g, h**). $P$ values were calculated by unpaired two-tailed Student's $t$ test (**c–e, g–h**). Uncropped immunoblots and statistical source data are provided in Source Data.

immune orchestration. If so, the next question would be which populations of immune cells are affected by CXCR2 ligands loss in Tbx3-deficient tumorigenesis. To answer this question, tumors from mPTC/Tbx3$^{-/-}$ and control littermates were subjected to FACS array with 7 immune lineage markers. Although macrophages, neutrophils, and MDSCs principally respond to CXCR2 ligands and play critical role in tumor immune evasion, MDSCs content was dramatically reduced in mPTC/Tbx3$^{-/-}$ tumors compared with control (Fig. 6a and Supplementary Fig. 6a-c). Regarding MDSC subpopulations, granulocytic MDSCs (G-MDSCs) were mainly decreased, while no significant difference was detected in the numbers of monocytic-MDSCs (M-MDSCs) (Fig. 6b). In contrast, the percentage of T cells, particularly CD8$^+$ T cells, and resultant CD8$^+$ T cell: G-MDSCs ratio was significantly augmented (Fig. 6c and Supplementary Fig. 6d). IHC staining further confirmed the lack of G-MDSCs (positive for Gr-1 and S100A8) and conspicuous infiltration of T cells in mPTC/Tbx3$^{-/-}$ (Fig. 6d)[40]. Similarly, in tumor tissues from mPTC-TAM/Tbx3$^{-/-}$ mutants, we detected decreased infiltration of MDSCs and increased ratio of CD8$^+$ T cells: G-MDSCs (Fig. 6e, f and Supplementary Fig. 6e, f). Co-culture assay showed that G-MDSCs from mPTC were capable of suppressing CD8$^+$ T cell proliferation, which was hardly affected by Tbx3 knock-out (Supplementary Fig. 6g). Thus, the augmented tumor-infiltrated cytotoxic CD8$^+$IFNγ$^+$GZM-B$^+$ T cell content is more due to decreased infiltration of G-MDSCs (Supplementary Fig. 6h). To further evaluate the anti-tumor effect of CD8$^+$ T cell in Braf$^{V600E}$-induced mPTC, we depleted CD8$^+$ T cell with anti-CD8 antibody. Remarkably, the anti-tumor effects of CD8$^+$ T cells were abrogated specifically in mPTC/Tbx3$^{-/-}$ with recovered tumor growth, since cytotoxic CD8$^+$ T cell were significantly depleted compared to anti-IgG in mPTC/Tbx3$^{-/-}$ (Fig. 6g and Supplementary Fig. 6i). These results demonstrate that Tbx3 is required for orchestration of immune-suppressive TME, knock-out of which cascade favors immune-cytotoxicity.

Interestingly, circulation and splenic G-MDSCs abundance were also decreased in mTPC/Tbx3$^{-/-}$, while bone marrow MDSCs and granulocyte-monocyte progenitors (GMPs) remain constant (Supplementary Fig. 7a, b). Considering the intermediary function of ligands-CXCR2 interaction between cell interaction, we wondered whether compromised MDSCs infiltration in mutants is due to CXCR2 ligands reduction caused attraction deficit. Indeed, the secretion of CXCL1 and CXCL2 was significantly suppressed in mTPC/Tbx3$^{-/-}$, concurrent with reduced serum CXCL1 and CXCL2 levels (Supplementary Fig. 7c). Whereas, we detected abundant and stable expression of CXCR2 on MDSCs (Supplementary Fig. 7d). As expected, conditioned media of primary mPTC tumor cells attracted MDSCs in chemotaxis assay, which was dramatically diminished by Tbx3 removal (Supplementary Fig. 7e). While chemical interruption of CXCR2-ligands communication by SB265610 repressed tumor cell-induced MDSCs migration significantly (Supplementary Fig. 7e). Similarly, conditioned medium from TBX3 knock-

down PTC cells showed impaired neutrophil attraction, which was rescued by over-expressed CXCR2 ligands (Supplementary Fig. 7f). SB265610 treatment significantly suppressed recruitment of neutrophils as well (Supplementary Fig. 7g). Together, TBX3-CXCLs axis induces chemotaxis of MDSCs and neutrophils through CXCR2-dependent paracrine, which also supports the rational that G-MDSCs represent pathologically activated neutrophils.

We further evaluated the impact of CXCR2-ligands interruption on mPTC progression in vivo. Consistent with clinical findings, PLX4032 reduced the tumor volume efficiently, and SB265610 alone also showed effective repression of tumor growth. Excitingly, combined PLX4032 and SB265610 treatment impeded tumor progression synergistically (Fig. 6h). At the cellular level, MDSCs content dropped down to different extents after treatments, and ratios of CD8$^+$ T cell: MDSCs were elevated (Fig. 6i, j and Supplementary Fig. 7h). Collectively, our findings indicate that TBX3-CXCLs axis, via binding to CXCR2, promotes MDSCs recruitment to build up immune-suppressive TME. Antagonizing the pathological events will complement the therapy effect of BRAF/MAPK pathway targeting.

**MDSCs mediate BRAF$^{V600E}$-induced immune-suppression and promote PTC advancement.** Given the critical function of TBX3 during BRAF$^{V600E}$-induced tumorigenesis and the direct regulation on MDSCs function, we are curious about the general immune environment in PTC development. Analyzed through FACS array, we found that the development of PTC was accompanied with intense immune environment reconstitution, especially infiltration of myeloid-derived cells, antigen-presenting cells, and tumor killer cells (Fig. 7a and Supplementary Fig. 8a). With PTC progression, infiltration of myeloid cells, especially G-MDSCs, the major subpopulation of MDSCs in PTC, was dramatically increased (Fig. 7b, c). FlowJo analysis further revealed that T cell contents, particularly CD8$^+$ T cells decreased significantly. In comparison, the percentages of DC cells declined gently, while tumor-associated macrophages (TAM) and natural killer cells (NK cells) showed little variations. Thus, the ratio of CD8$^+$ T cells: G-MDSCs decreased significantly (Fig. 7d). Molecular marker staining also confirmed elevated infiltration of MDSCs while declined T cell abundance (Fig. 7e). Next, thyroid tumor tissues were harvested from mPTC-TAM mice at different stages and subjected to FACS array. As we observed in mPTC, percentage of MDSCs, especially G-MDSCs, gradually increased with the tumor progression, while T cells decreased (Fig. 7f-h and Supplementary Fig. 8b). Since this is a gradually progressed tumor model, the infiltration of immune cells differs from mPTC model so that the infiltration of macrophages and DC cells also went up a bit. In clinical PTC samples, CD33 and S100A8 (markers of human MDSCs) showed much stronger signals than adjacent normal tissues (Supplementary Fig. 8c). Together, high penetrance of MDSCs becomes representative phenomenon in PTC initiation and progression.

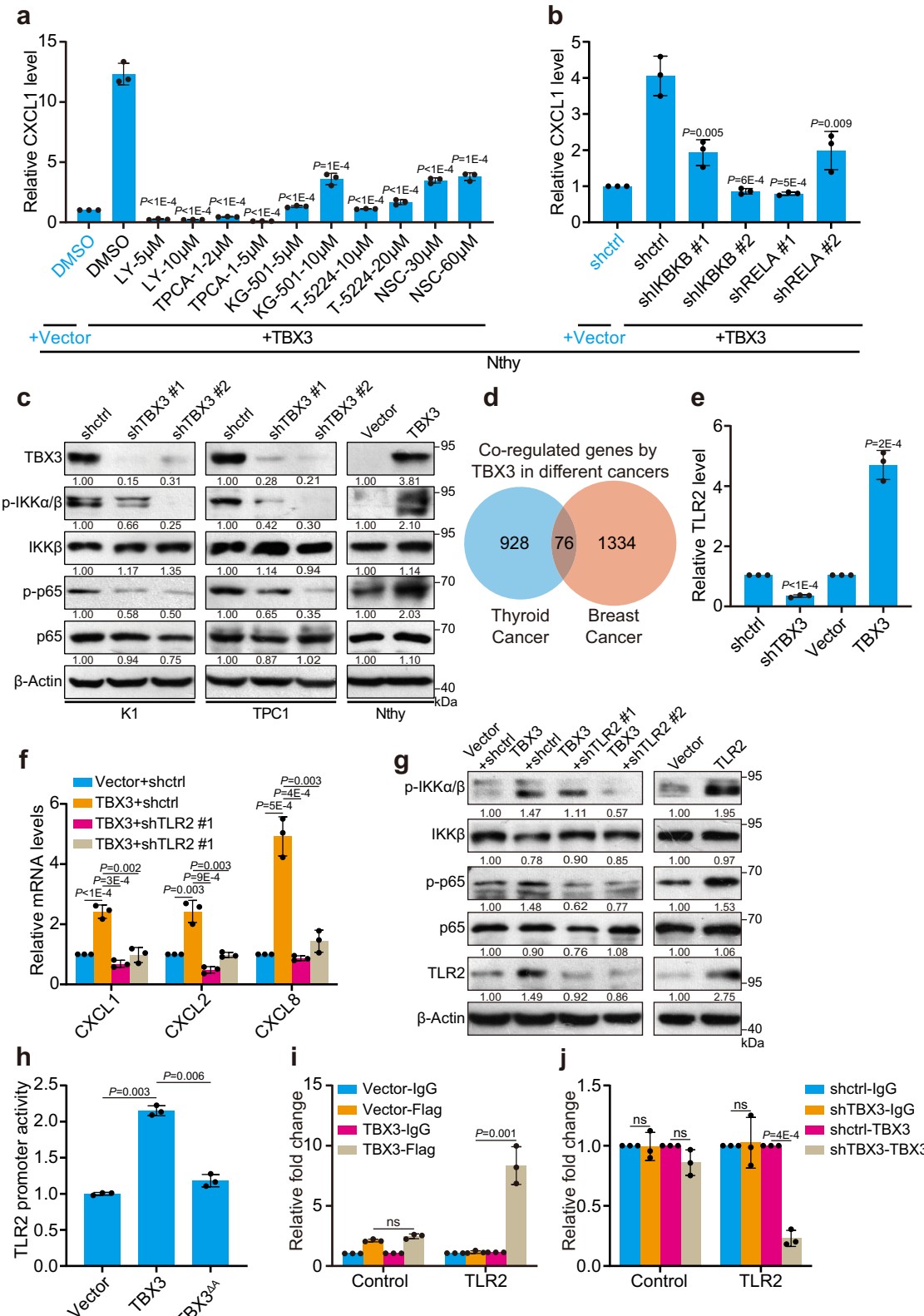

We next investigated whether antagonism of MDSCs with anti-Gr-1 antibody would affect PTC development. After 4 weeks of treatment, the mPTC tumor weight was reduced to about 70–80% of the initial weight by anti-Gr-1, while combined anti-Gr-1 antibody and PLX4032 showed synergistic effect and shrank the tumors more effectively (Fig. 7i). Through quantification analysis,

MDSCs were restricted, so that the ratio of CD8+ T cells: MDSCs was elevated (Fig. 7j). It is worth noting that single-agent therapy with PLX4032 also reduced the infiltration of MDSCs while increased that of CD8+ T cells. Histological analysis also revealed the reduction of MDSCs in anti-Gr-1 treated tumors, which was further reduced through PLX4032 single and combined therapy.

**Fig. 4 TBX3/TLR2-NF-κB axis induces chemokines expression. a** CXCL1 mRNA level in TBX3 over-expressed Nthy cells treated with small-molecule inhibitors against IKKβ (LY2409881 (LY), TPCA-1), CREB (KG-501), AP-1 (T-5224) or STAT3 (NSC74859 (NSC)) was measured by RT-qPCR, comparison between DMSO- and inhibitor-treated TBX3-over-expressed Nthy cells was used for statistical analyses. **b** RT-qPCR analysis of CXCL1 in TBX3 over-expressed Nthy cells infected with shRNAs against IKBKB or RELA, comparison between shctrl- and shRNA- TBX3-over-expressed Nthy cells was used for statistical analyses. **c** Western blot of NF-κB pathway members in K1, TPC1 with TBX3 knock-down, and Nthy cells with TBX3 over-expression. **d** Venn diagram analysis across two groups of genes. Down-regulated, shTBX3 versus control in K1 cells ($n = 1004$); Up-regulated, TBX3 over-expression versus control in 21NT breast cancer cells ($n = 1410$). **e** RT-qPCR analysis of TLR2 in K1 cells with TBX3 knock-down or over-expression. **f** RT-qPCR analysis of CXCL1, 2 and CXCL8 in TBX3 over-expressed K1 infected with shRNAs against TLR2. **g** Western blot of NF-κB pathway members in K1 cells as in (**f**). **h** HEK293T cells were co-transfected with TLR2-GLuc, TBX3 and TBX3 with activator domain deletion (TBX3$^{ΔA}$) assayed for GLuc activity using SEAP activity as control. **i** qChIP analysis using anti-IgG and anti-FLAG antibody was performed in K1 cells with Flag-TBX3 over-expression. **j** qChIP analysis using anti-IgG and anti-TBX3 antibody was performed in K1 cells with TBX3 knock-down. Densitometric analyses of western blot were shown (**c, g**). $n = 3$ biological independent samples (**a, b, e–j**), and two independent experiments were carried out with similar results for each kind of cells (**c**). Data are shown as the mean ± s.d. (**a, b, e, f, h–j**). $P$ values were calculated by unpaired two-tailed Student's $t$ test (**a, b, e, f, h, j**). Uncropped immunoblots and statistical source data are provided in Source Data.

Resultantly, more CD4$^+$ and CD8$^+$ T cells were detected into the TME (Fig. 7k). Together, these data reinforce our finding that MDSCs regulated by oncogenic pathways contribute to PTC progression, and that blocking MDSCs function could enhance therapy efficacy of MAPKi.

**TBX3-CXCLs regulation transduces BRAF$^{V600E}$-induced MDSCs recruitment.** Based on the above findings, we hypothesized that TBX3-CXCLs could be one of the mechanisms linking BRAF$^{V600E}$-induced transformation and MDSCs recruitment. First, we determined whether CXCLs were under BRAF/MAPK pathway control and subjected mPTC$^{GFP}$ cells to transient MAPKi treatment. Gene expression analysis was followed under quality control with p-ERK1/2 level, where multiple Cxcls were significantly decreased by BRAF/MAPK inhibition (Supplementary Fig. 9a).

Indeed, CXCR2 ligands were increased during mPTC development at both mRNA and protein levels, but decreased after PLX4032 therapy (Supplementary Fig. 9b, c). Similarly, CXCR2 ligands were highly expressed in PTC patient and down-regulated by MAPKi in PTC cells (Supplementary Fig. 9d, e and Fig. 8a). We further found that TBX3 is required for BRAF$^{V600E}$-induced expression of these CXCR2 ligands (Fig. 8b). Therefore, TBX3-promoted CXCR2 ligands high expression could function downstream of BRAF/MAPK activation and participate in immune environment regulation.

To gain a more general insight into the association between TBX3, chemokines, and immune cell infiltration in clinical specimens, we performed IHC on 110 PTC specimens. Notably, IHC analysis using Image-Pro Plus showed that expression of TBX3 and CXCLs were concurrently up-regulated and correlated with histological grades (Fig. 8c, d). Moreover, TBX3 levels correlate positively with the expression of CXCL1, CXCL2, as well as CXCL8 respectively ($p < 0.0001$) (Supplementary Fig. 9f). Further evaluation indicated that TBX3 level in BRAF$^{V600E}$ positive samples was statistically higher than that in mutation-negative samples, which confirmed the clinical relevance of our findings (Fig. 8e). Consequently, the levels of CXCL1 and CXCL8 were significantly correlated with BRAF$^{V600E}$ mutation, although CXCL2 was only gently increased in BRAF$^{V600E}$ tissues (Fig. 8e). Taken together, these results demonstrate that TBX3-directed CXCR2 ligands elevation causes increased MDSCs infiltration. This is a critical molecular cascade during BRAF$^{V600E}$-induced PTC development that could be a promising therapy target adjuvant with MAPKi (Fig. 8f).

**Discussion**
By attempting to understand the global landscape of advanced PTC immune microenvironment, here we delineate a function of

oncogenic BRAF$^{V600E}$, which drives an immune-suppressive program by recruiting MDSCs. Transcriptional re-activation of TBX3 is an indispensable molecular event for cancer initiation and progression, which linked BRAF/MAPK pathway activation and CXCR2 ligands-attracted MDSCs infiltration. Through exploring critical molecular mechanisms, we provide a rationale for developing therapeutic approaches combining inhibition of internal oncogenic events and subversion of tumor immune-suppression. Moreover, in proof-of-concept studies, we demonstrate that inhibition of MDSCs recruitment or activity through antagonizing CXCR2 or Gr-1 improves the therapeutic effect of BRAF inhibitor.

Studies about infiltration and function of innate immune cells, mostly DC cells, macrophages, neutrophils, and NK cells, are contradictory in TC, making it elusive to simply link them with tumor progression[11]. But the density of lymphocytes, especially CD4$^+$, CD8$^+$ T cells, correlated with improved overall survival and lower recurrence[41]. Studies on the role of MDSCs, the major driver of immune-suppressive environment, in TC progression was limited to the reported increase in peripheral blood mononuclear cells of PTC patients[19,20]. In the current study, we found that MDSCs infiltration is gradually ascending during tumor progression, which negatively correlates with CD8$^+$ T cell abundance in both genetic PTC model and clinical specimens. MDSCs antagonize activation and proliferation of T cells through secreting ROS and other cytokines, or there is also evidence that MDSCs inhibit recognition of tumor cell associated MHC and cause resistance of tumor cells to cytotoxic T cells[42]. Considering that MHCI and MHCII are both repressed on advanced PTC cells by constitutively activated BRAF/MAPK pathway, we could speculate that oncogenic pathways promote immune escape through coordinating multiple tumor and stromal cell behaviors[12,43]. Therefore, the infiltration of MDSCs may serve as a potential biomarker for PTC diagnosis stratification and prognosis pre-warning.

Our findings bear clinical implications. First, it explains why PLX4032 treatment alone led to compromised infiltration of MDSCs, which partially accounts for the improved therapy effect of PLX4032 with anti-PD-L1 antibody combined therapy[12,44]. In addition, either CXCR2 antagonist or Gr-1 blockage shifted the ratio between MDSCs and CD8$^+$ T cells, and improved the therapeutic effect of PLX4032, which confirms that this cell population could be a potential therapy target[45,46]. Therefore, synergistic repression of tumor cell proliferation and reversion of immune-suppression could represent one of the most effective strategies to control PTC advancing and relapse.

Within tumor cells, TBX3 is one critical factor initiated by BRAF$^{V600E}$-induced oncogenic signalings and participates in immune environment orchestration. Aberrant *TBX3* gene

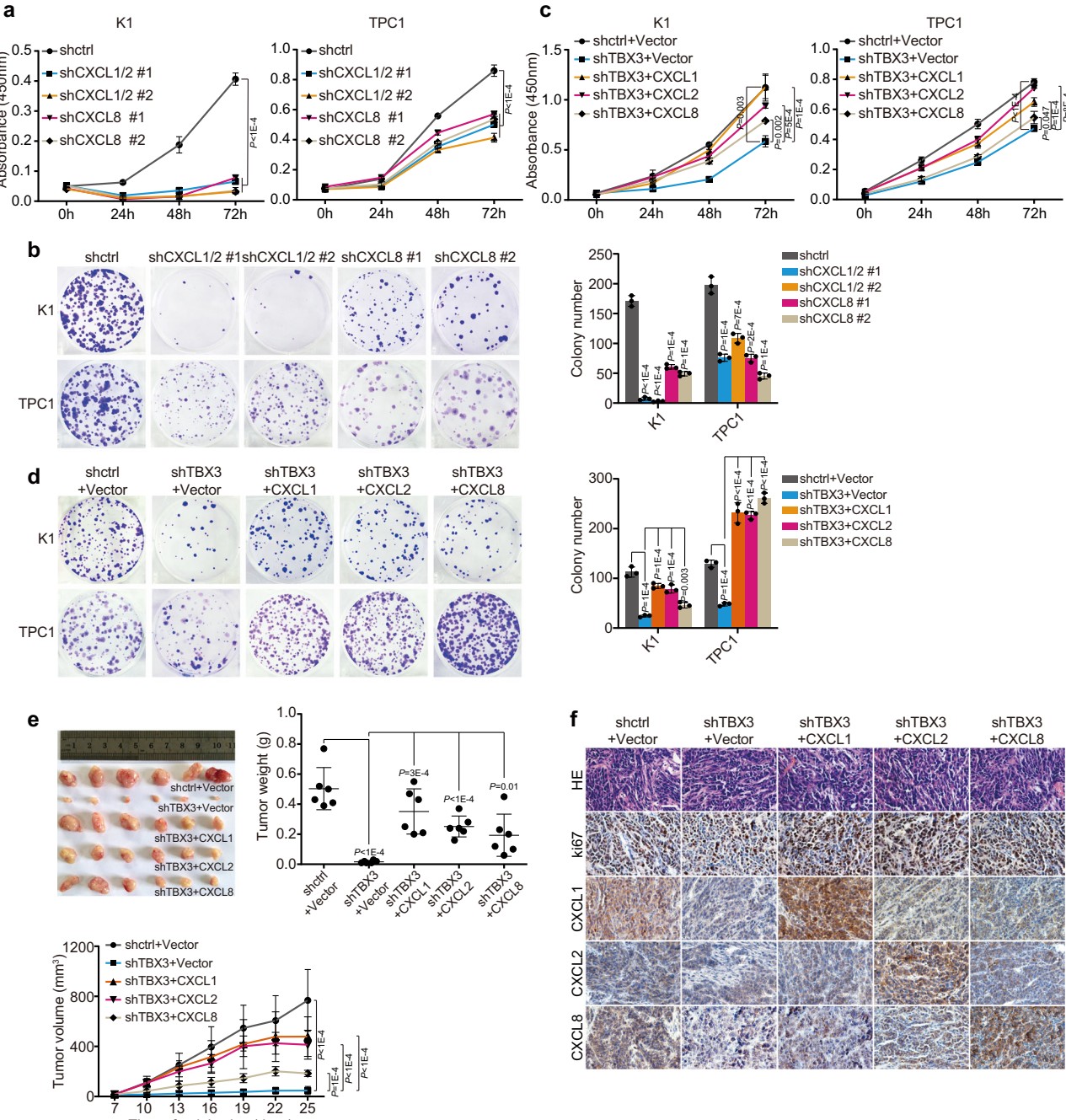

**Fig. 5 CXCR2 ligands function downstream of TBX3 and promote tumor cell proliferation. a, b** PTC cells were infected with specific shRNA against CXCL1/2 or CXCL8, and subjected to CCK8 (**a**) or colony formation assay (**b**). **c, d** TBX3 knock-down PTC cells with or without CXCLs over-expression were subjected to cell growth assays of CCK8 (**c**) or colony formation (**d**). **e** K1 cells with TBX3 knock-down and Vector or CXCLs over-expression were transplanted into athymic mice subcutaneously. Tumor weights as well as curves were generated and statistically compared, $n = 6$. **f** HE and IHC staining of indicated factors were performed on tumor sections in (**e**). Scale bars, 50μm. $n = 3$ biological independent samples (**a–d**). Data are shown as the mean ± s.d. (**a–e**). $P$ values were calculated by unpaired two-tailed Student's $t$ test (**a–e**). Statistical source data are provided in Source Data.

expression has been reported in several malignancies, but upstream regulators and many of the downstream targets are still not defined. During embryonic or postnatal oncogenic transformation, we observed progressively up-regulated Tbx3 expression that is otherwise barely detectable in adult thyroid. Failed activation of Tbx3 by genetic ablation dramatically prevents the tumor initiation and ameliorates BRAF[V600E]-associated reduction of lineage factors, indicating that re-opening of a primitive

developmental factor could be a fundamental event for local tumorigenesis. Whether TBX3 is downstream of BRAF/MAPK signaling during early development remains to be further investigated. As an indispensable factor required for mammary development, TBX3 is also highly involved in breast cancer[47]. Molecularly, AP-1 transcriptional factors, as the principal mediator downstream of BRAF/MAPK cascade, directly bind to -149 AP-1 site within *TBX3* gene regulatory region and activate the

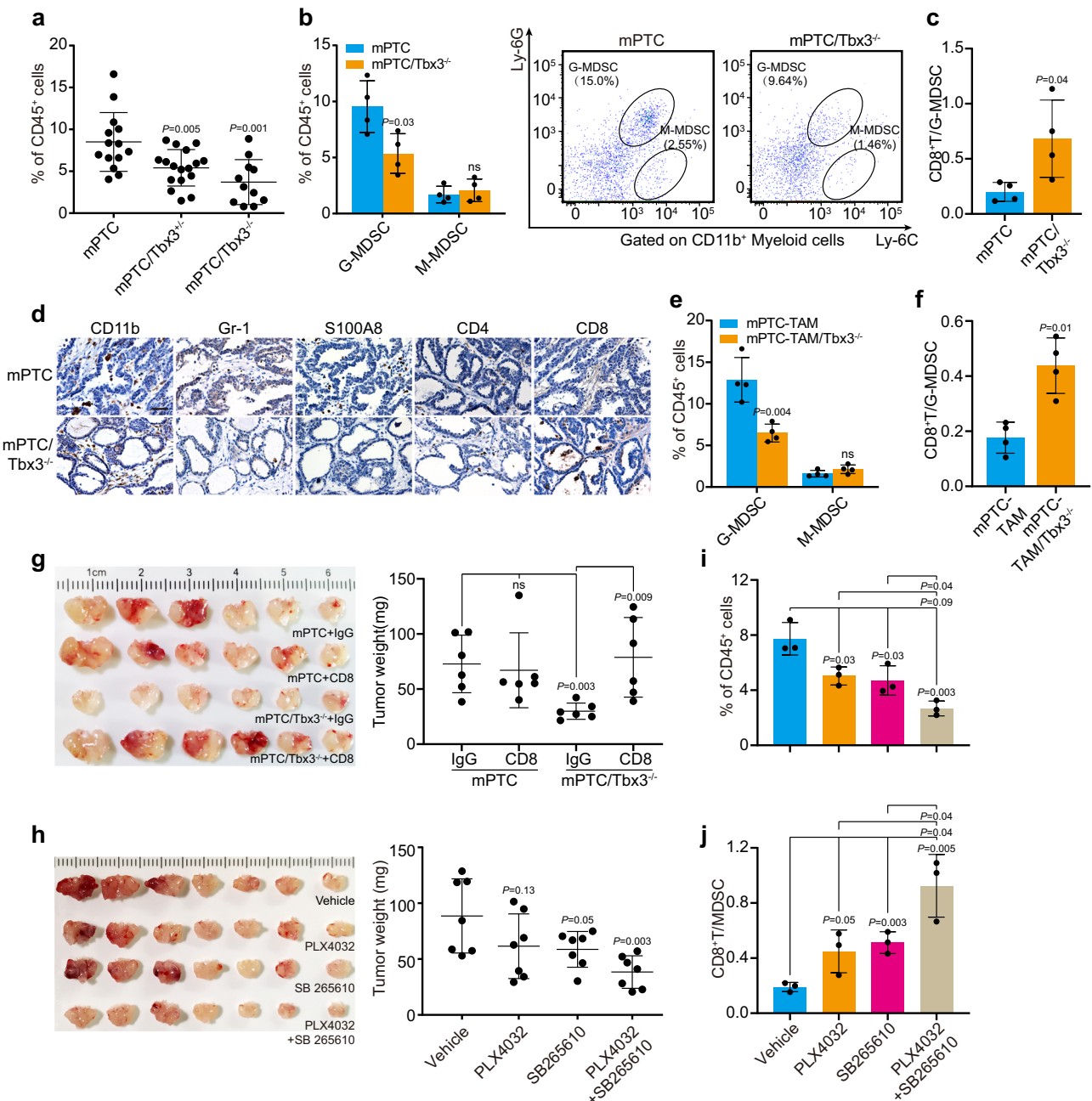

**Fig. 6 TBX3 deficiency blocked tumor formation due to suppressed CXCR2-mediated MDSCs recruitment. a** Flow cytometric analysis of MDSCs (CD45+CD11b+Gr-1+) in thyroid glands from mPTC (*n* = 14), mPTC/Tbx3+/− (*n* = 17) or mPTC/Tbx3−/− (*n* = 11) littermates at 5w. **b, c** Percentage of G-MDSC (CD45+CD11b+Ly6C-Ly6G+) and M-MDSC (CD45+CD11b+Ly6C+Ly6G-) of CD45+ tumor-infiltrating leukocytes (TILs) in mPTC tumors compared with mPTC/Tbx3−/− littermates at (**b**). Relative CD8+ T cells/MDSCs was plotted (**c**), *n* = 4. **d** IHC analysis of CD11b, MDSCs (Gr-1, S100A8), CD4+ T cells (CD4) and CD8+ T cells (CD8) in thyroid glands from mPTC/Tbx3−/− compared with mPTC littermates. Scale bars, 50μm. **e, f** Flow cytometric analysis of G-MDSC and M-MDSC in mPTC-TAM or mPTC-TAM/Tbx3−/− tumors induced with tamoxifen for 8 m. CD8+ T cells/MDSCs was plotted, *n* = 4. **g** Mice bearing mPTC and mPTC/Tbx3−/− tumors were treated with 200 μg anti-CD8 antibody per mouse twice weekly at age of 3w for 20d. Tumor weights were counted, *n* = 6. **h** Mice bearing tumors received daily oral doses of PLX4032 10 mg/kg body weight or SB265610 2 mg/kg body weight at age of 3w for 20d. Tumor weights were counted, *n* = 7. **i, j** Flow cytometric analysis of mPTC tumors underwent above treatments. Percentage of positive cells relative to total cells was plotted. MDSCs (**i**) and CD8+ T cells/MDSCs (**j**), *n* = 3. A representative of three independent experiments was shown (**d**). Data are shown as the mean ± s.d. (**a–c**, **e–j**). *P* values were calculated by unpaired two-tailed Student's *t* test (**a–c**, **e–j**). Statistical source data are provided in Source Data.

transcription. In comparison, c-Jun specifically binds with -371 (here termed as -360) AP-1 site and transactivates *TBX3* expression during breast cancer development[48]. The preference for different AP-1 sites may relate very much to the epigenetic modification and chromatin state under each tumor initiation.

Clinically, TBX3 is highly expressed in BRAF^V600E mutation-positive PTC samples, and the transcriptional activation seems to be playing the major role since the coding-gene mutations and epigenetic de-repression occur in PTC with low frequency. Nowadays, the overtreatment of thyroid cancers also becomes a

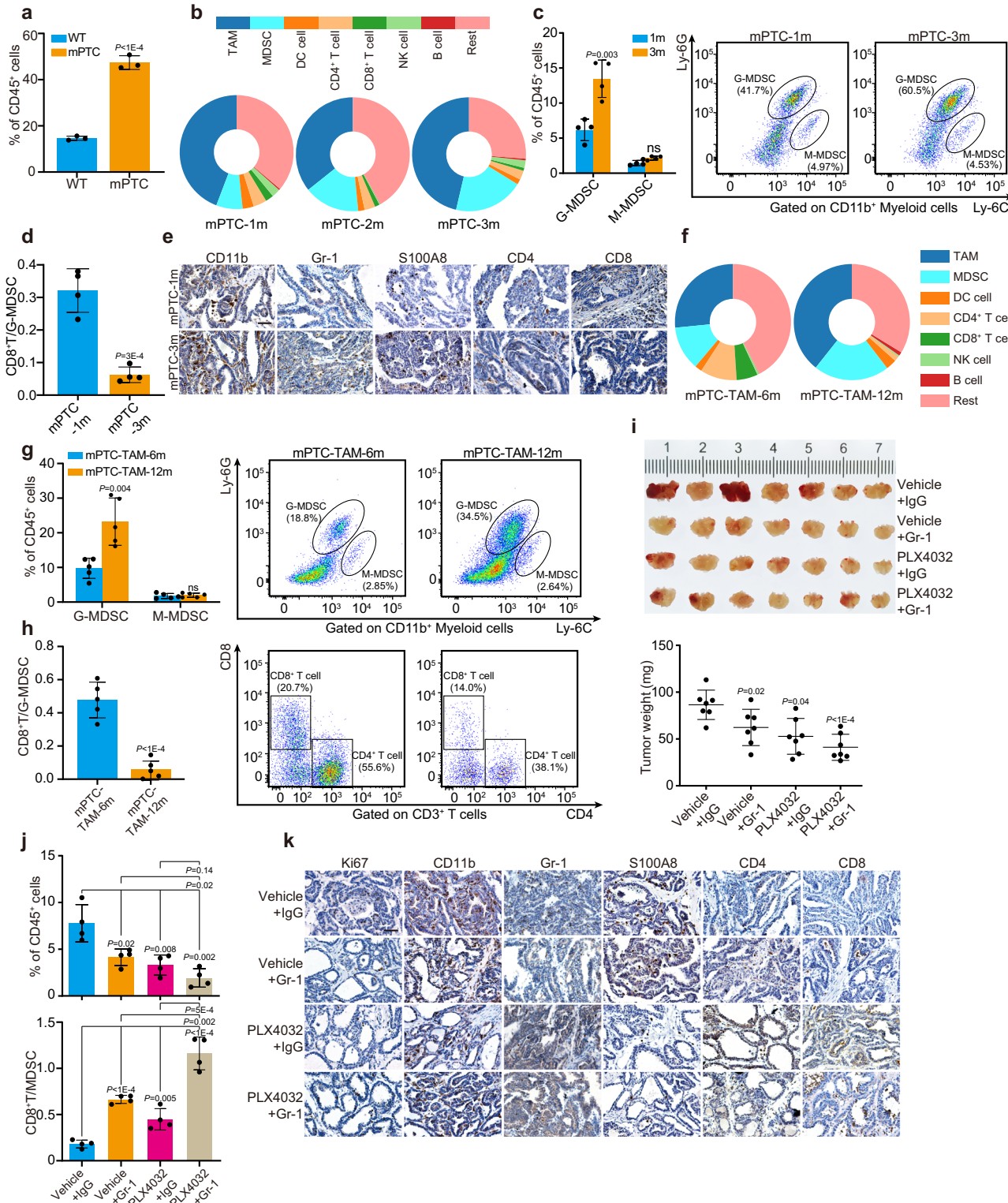

medical care problem. Through validation among a larger number of PTC cases, TBX3 will be a promising biomarker candidate in terms of diagnosis stratification, prognosis prediction, and risk warning in conjunction with BRAF[V600E]. However, whether TBX3 level correlates with other genetic abnormity in PTC advancing needs further investigation.

CXCR2 ligands are especially important for MDSCs recruitment into tumor stroma or pre-metastatic niche[49,50]. RNA-seq and

antibody screening suggested several ligands are predominantly increased during BRAF/MAPK activation-induced tumorigenesis, among which CXCL1, 2, and 8 are indeed up-regulated in PTC specimens. It is not surprising that several CXCR2 ligands were synergistically regulated within TME. For example, in colorectal cancer, CXCL1, 2, and 5 are parallelly increased in tumorigenesis[42]. Previous study shows that CXCL8 is increased in patient serum with advanced thyroid carcinoma[51]. Similarly, in gastric, colorectal, and

**Fig. 7 Antagonizing MDSCs inhibits Braf$^{V600E}$-induced immune-suppression and tumor advancement. a** Percentage of tumor-infiltrating myeloid immune cells (CD45$^+$CD11b$^+$) of CD45$^+$ TILs in mPTC at 5w compared with wild-type littermates, analyzed by FlowJo, $n = 3$. **b** Quantification of whole leukocyte populations in thyroid glands of mPTC at 1 m, 2 m and 3 m by FlowJo, $n = 4$. **c** Percentage of G-MDSCs and M-MDSCs of CD45$^+$ TILs in mPTC tumors at 1 m and 3 m, $n = 4$. **d** Relative CD8$^+$ T cells/G-MDSCs of mPTC at 1 m or 3 m was plotted, $n = 4$. **e** IHC analysis of CD11b, Gr-1, S100A8, CD4 and CD8. Scale bars, 50μm. **f** Quantification of whole leukocyte populations in thyroid glands of mPTC-TAM induced with tamoxifen for 6 m and 12 m, $n = 5$. **g, h** FlowJo analysis of G-MDSCs, M-MDSCs, CD4$^+$ T cells and CD8$^+$ T cells of CD45$^+$ TILs in mPTC-TAM tumors induced with tamoxifen for 6 m and 12 m (**g**), CD8$^+$ T cells/G-MDSCs was plotted (**h**), $n = 5$. **i** Mice bearing tumors were treated with anti-IgG antibody or anti-Gr-1 antibody three times a week at age of 3w for 20d, and received daily oral doses of PLX4032 or vehicle for 20d. Tumor volumes and tumor weights were counted, $n = 7$. **j** Flow cytometric analysis of tumors underwent above treatments. Percentage of positive cells relative to total cells was plotted. MDSCs (upper) and CD8$^+$ T cells/MDSCs (lower), $n = 4$. **k** IHC staining in four groups of mPTC from (**i**). Scale bars, 50μm. A representative of three independent experiments was shown (**e, k**). Data are shown as the mean ± s.d. (**a, c, d, g–j**). $P$ values were calculated by unpaired two-tailed Student's $t$ test (**a, c, d, g–j**). Statistical source data are provided in Source Data.

hepatocellular carcinoma serum CXCL1 and CXCL2 correlate with tumor size, metastasis, and decreased overall survival[52–54]. Therefore, local CXCR2 ligands promote tumor proliferation and recruit MDSCs through autocrine or paracrine.

In human PTC study, up-regulated TBX3 correlates with high MDSCs infiltration, which happens more often in BRAF$^{V600E}$ mutation-positive patients. It's worthwhile to analyze TBX3 level and MDSCs signature in relapsed or RAI-refractory patient samples that respond differently to single MAPKi treatments. Because of the functional antagonism and abundance negative correlation between MDSCs and CD8$^+$ T cells, the involved patients may not respond well to anti-PD-1 therapy. In these cases, immune therapy against MDSCs combined with MAPKi drugs could be a potential therapy strategy. Therefore, TBX3 could be a valuable reference for establishing customized therapy plans. In terms of etiology, whether TBX3 high expression correlates with high PTC relapse and treatment-refraction is also an important issue to address to consider it as a prognosis prediction marker. In summary, our findings uncover a critical role for BRAF/TBX3/CXCLs signaling in the regulation of anti-tumor immunity and suggest the use of combination CXCR2 inhibitor with MAPKi therapy in patients with advanced PTC.

## Methods

**Ethic approval**. For clinical samples and human blood, the patients and healthy volunteers all signed an informed consent form issued by the Tianjin Cancer Institute and Hospital Ethics Committee, and the study was approved by the Ethics Committee. All mouse experiment procedures and protocols were evaluated and authorized by the Regulations of Tianjin Laboratory Animal Management and followed the guidelines under the Institutional Animal Care and Use Committee of Tianjin Medical University (Approval number: SKLAB-2018-04-03).

**PTC samples from patients**. All PTC samples were obtained from Tianjin cancer institute and hospital (Tianjin, China). Fresh samples from adjacent normal tissues, pathological grade I, II, III, and IV were frozen in liquid nitrogen immediately after resection or fixed in 4% paraformaldehyde (Sigma Aldrich) at 4 °C overnight before embedded in paraffin. Patients with a past history of radiation therapy or chemotherapy were excluded from this study. Samples were then used for mRNA extraction or immunohistochemistry according to standard protocols. To analyze BRAF$^{V600E}$ mutation, PCR was performed with tissue cDNA and used for direct DNA sequencing. The primers were as follows: forward primer: 5′-TCGAGT-GATGATTGGGAGATTC-3′ and reverse primer: 5′-CTCAATAGAGGCGA-GAATTTGG-3′.

**Mice**. Animals used in this study were maintained as specific-pathogen free (SPF) mice. *TPO-Cre* mice were gifted by Professor Shioko Kimura of NIH, USA[55], *LSL-Braf$^{V600E}$CA* mice were obtained from Professor Martin McMahon of NIH, USA[56], and *Tbx3$^{flox/flox}$* mice and *Tbx3-GFP* mice were gifted by Professor Anne Moon of University of Utah[32], *TPO-creER* mice and *Rosa-mTmG* mice were purchased from The Jackson Laboratory, USA. Expression of Cre recombinase under control of the thyroid-specific TPO promoter[57], leads to the expression of Braf$^{V600E}$, GFP or Tbx3$^{G/+}$, as well as deletion of Tbx3 in *TPO-cre/Braf$^{V600E}$* (mPTC), *TPO-cre/Braf$^{V600E}$/mTmG* (mPTC$^{GFP}$) *TPO-cre/Braf$^{V600E}$/Tbx3$^{GFP/+}$* (mPTC/Tbx3$^{G/+}$) or *TPO-cre/Braf$^{V600E}$/Tbx3$^{flox/flox}$* (mPTC/Tbx3$^{−/−}$) line, respectively. Braf$^{V600E}$-induced thyroid tumors and mouse blood were collected at 5w to analyze the development of tumors and levels of T3, T4, or TSH in serum. For PTC in the adult

mouse, *TPO-creER* mice were bred to *LSL-Braf$^{V600E}$CA* to generate *TPO-creER/Braf$^{V600E}$* mice, in which activation of Braf$^{V600E}$ was achieved by intraperitoneal injection of 100 μl of a 10 mg/mL stock of tamoxifen dissolved into corn oil at age of 1 m (mPTC-TAM for tamoxifen treatment and mPTC-CON for oil treatment). Tumor tissues were obtained at 6 m, 8 m, 10 m, or 12 m to analyze tumor progression.

**Cell culture and stable cell line**. HEK293T (ACS-4500) was purchased from the American Type Culture Collection (ATCC). And all cancer cell lines were obtained from Tianjin Medical University Cancer Institute and Hospital with STR profiling, and cultured in Dulbecco's modified Eagle's medium (DMEM) supplemented with 10% fetal bovine serum (FBS) and 1% penicillin/streptomycin, while Nthy-ori 3-1 (Nthy) and BCPAP cultured in RPMI-1640 with 10% FBS, maintaining in a humidified incubator equilibrated with 5% CO$_2$ at 37 °C. For preparation of lentivirus, the core plasmids with lentivirus construction were co-transfected into HEK293T cells with packing plasmids by lipofectamine 3000 (#L3000015, Invitrogen). The virus particles were harvested and filtered by 0.45 μM filter unit (Millipore). Tumor cells were then infected with lentivirus and screened for stable cell lines with presence of 1 μg/ml puromycin. The cDNAs of TBX3, BRAF, BRAF$^{V600E}$, c-Jun, JunB, c-Fos, CXCL1, CXCL2, CXCL8 and TLR2 were cloned into lentiviruses construction pCDH-CMV-puro. Recombinant lentiviruses expressing different shRNAs were obtained by cloning designed shRNA into pLKO.1-puro. The shRNA sequences are listed in Supplementary Table 1. Crispr system targeting TBX3 was performed according to our previous study[24].

**Cell proliferation and colony formation assays**. For cell proliferation assay, K1 or TPC1 cells were seeded in triplicate into 96-well plates with 2000 cells per well. The cell proliferation index was measured using the Cell Counting Kit-8 (#CK04-500, Dojindo) at different times according to the manufacturer's instruction at a wavelength of 450 nm. For colony formation assay, K1 or TPC1 cells were seeded in triplicate into 6-well plates at a density of 500 cells per well and maintained in DMEM medium containing 10% FBS for 2 w. The total number of colonies (Diameter ≥ 100 μm) was counted and representative areas were imaged under a microscope after staining with 0.5% crystal violet (#C6158, Sigma).

**Neutrophil chemotaxis assays**. Neutrophils were isolated from the blood of healthy volunteers. About 5 ml blood was used to harvest the neutrophils with percoll and cultured in RPMI-1640 with 10% FBS[58]. The purity of neutrophils was >97% as assessed by Romanowski staining. Neutrophil chemotaxis assays were performed in triplicates using Transwell (3.0 μM, Corning) in 24-well plates. Approximately 5 × 10$^5$ neutrophils were plated in the upper chambers with 200 μl RPMI-1640 per well and the medium from tumor cells with knock-down of TBX3 was placed in the lower chamber. After incubation for 2 h, the number of neutrophils in the bottom compartment was counted.

**Animal experiments**. For xenograft tumorigenicity, K1 cells with TBX3 knock-down and/or CXCLs over-expression were injected subcutaneously into the dorsal flanks of the female mice ($n = 8$) with a total of 5 × 10$^6$ cells/150 μl PBS per BALB/c nude mouse at age of 5–6 w, which were purchased from Beijing Vital River Laboratory Animal Technology Company (Beijing, China). Tumors nod length (L) and width (W) were measured every 3 days using a vernier caliper, and the volume was calculated according to $1/2LW^2$. Mice were euthanized while the smallest tumor was detectable and the largest tumor was strictly under 2000 mm$^3$ according to the ethics committee requirements. Paraffin embedding and immunohistochemical analysis were then carried out.

For therapy using mPTC model, mice bearing tumor at age of 3 w were randomized to several groups for further treatment. For antibody-based drug intervention, igG2a isotype control (#BE0089, Bio-XCell) and anti-Gr-1 (#BE0075, Bio-XCell) were given three times per week at a dosage of 200 μg/injection. For drug-based intervention, PLX4032 (#S1267, Selleck) was orally given as medicated chow at 10 mg/kg body weight, and SB265610 (#2724, R&D Systems) was

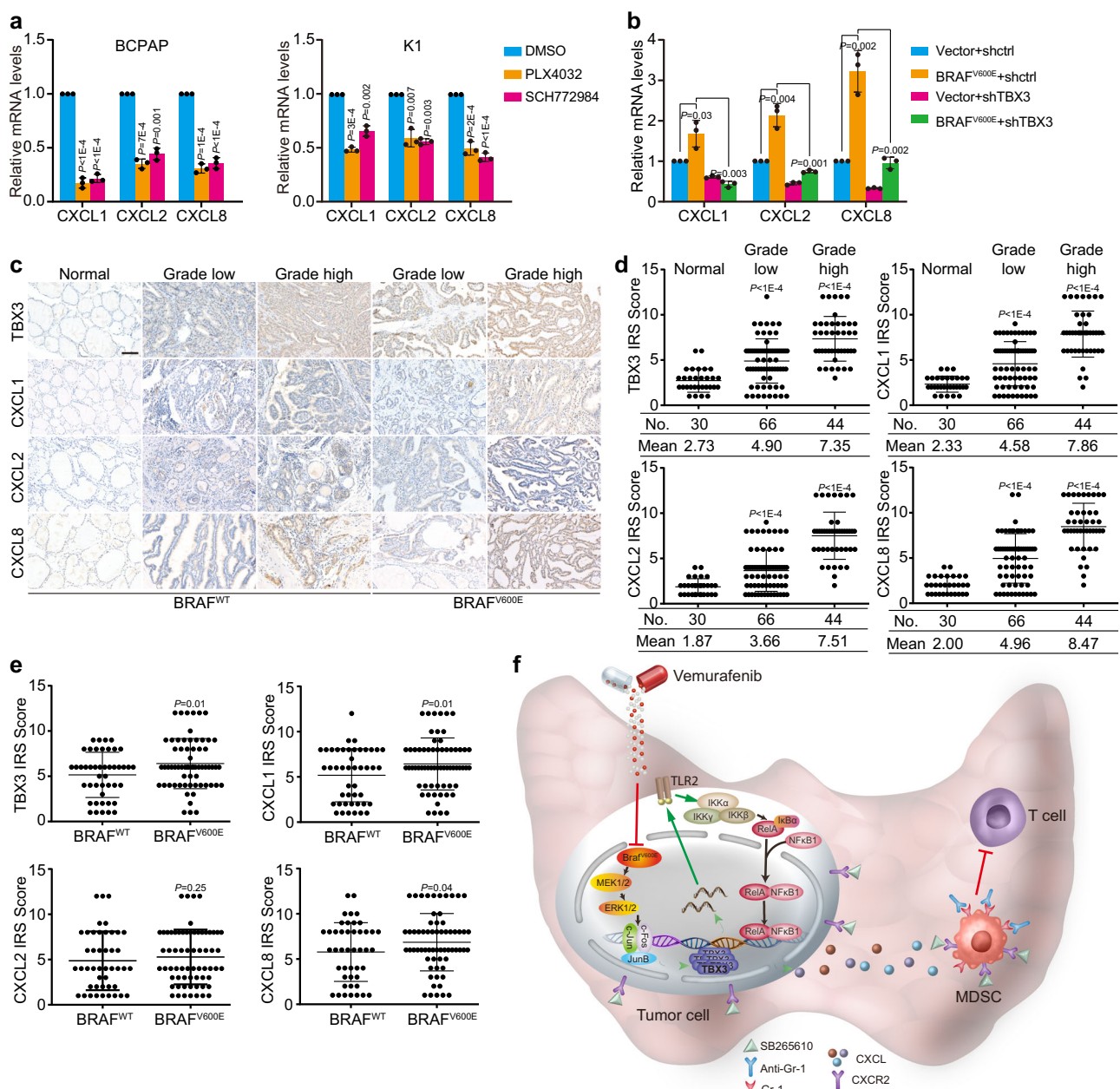

**Fig. 8 BRAF^V600E promotes CXCLs level and MDSCs infiltration through TBX3-regulated pathway. a** Levels of CXCLs in PTC cells underwent PLX4032 or SCH772984 treatment were checked by RT-qPCR. **b** Levels of CXCLs in K1 cells stably over-expressing control Vector or BRAF^V600E and subsequently infected with lentiviruses expressing TBX3 shRNAs were analyzed by RT-qPCR. **c**, **d** IHC staining in PTC samples with BRAF^WT or BRAF^V600E paired with adjacent normal tissues. Representative images were shown (**c**), and the scores of the stained sections were plotted (**d**), n = 30 normal tissues, n = 66 grade low PTC tissues, and n = 44 grade high PTC tissues. Scale bars, 100μm. **e** Scatter diagrams of TBX3, CXCL1, CXCL2 and CXCL8 protein levels in BRAF^WT (n = 45) and BRAF^V600E (n = 65) PTC tissues. **f** A proposed model of the mechanism of re-activation of TBX3 by BRAF/MAPK constitutive activation leading to immune-suppression by recruiting MDSCs into tumor microenvironment and the rational of combination therapy using PLX4032 and SB265610 or anti-Gr-1 to treat thyroid cancer. n = 3 biological independent samples (**a**, **b**). Data are shown as the mean ± s.d. (**a**, **b**, **d**, **e**). P values were calculated by unpaired two-tailed Student's t test (**a**, **b**, **d**, **e**). Statistical source data are provided in Source Data.

administered intraperitoneally six times a week with 2 mg/kg body weight, n = 7 per group. Mice bearing mPTC and mPTC/Tbx3^−/− tumors were treated with 200 μg anti-CD8 antibody (#BE0061, Bio-XCell) and IgG2a control per mouse twice weekly at age of 3 w for 20 d. Mice were sacrificed and isolated tumors were used for flow cytometry analysis or immunohistochemical analysis.

**Primary cell culture.** Cancer tissues obtained from mPTC^GFP were minced in lysis buffer (Collagenase I (Sigma) 1 mg/ml; Dispase II (Invitrogen) 0.5 mg/ml in PBS) and shaked on an orbital shaker at 37 °C for 60 min. Dissociated tumor cells were filtered with 40 μm cell strainer (#352340, BD Falcon) and centrifugated at 500 g for 5 min, the pellet was resuspended in cold Ham's F-12 Nutrient Mix (#11765062, Gibco). The isolation of tumor cells was next performed on FACSAria Cell Sorter (Becton Dickinson) according to GFP signal, and positive cells were washed with PBS and then cultured in F-12 medium supplemented with 10% FBS, 1% penicillin/streptomycin, 5 mg/L transferrin (Sigma-Aldrich), 10 mg/L bovine insulin (Solarbio), 3.5 mg/L hydrocortisone (Sigma-Aldrich), 10 mg/L somatostatin (Sigma-Aldrich), 0.02 mg/L Gly-His-Lysacetate (Sigma-Aldrich) and 1 IU/L bovine thyroid stimulating hormone (TSH) (Sigma-Aldrich) related to the 6H medium at 37 °C[59]. Next, Primary cells treated with 10 μM PLX4032 or 2 μM SCH772984 were subjected to western blot and RT-qPCR analysis. Primary cells from mPTC mice were directly cultured in complete F-12 medium. Thyrocytes were confirmed by immunofluorescence staining of thyroglobulin and representative photos were taken by a fluorescence microscopy.

**MDSCs in vitro migration assay**. MDSCs were isolated from the spleens of PTC mice using a Mouse MDSC Isolation Kit (#130-094-538, Miltenyi Biotec) and plated in RPMI1640 supplemented with 10% FBS. Conditioned media (CM) from cultured primary mPTC cell lines: mPTC and mPTC/Tbx3$^{-/-}$ was collected and added to the bottom layer of the transwell (3.0 μM). MDSCs (1 × 10$^5$ cells/well) were seeded in the top chamber of the transwell, CM was at bottom chamber with CXCR2 inhibitor for 4 h incubation. Cells migrated to the bottom chamber were counted.

**T-cell proliferation assay**. MACS-sorted G-MDSCs from mPTC and mPTC/Tbx3$^{-/-}$ tumors were planted with CD8$^+$ T cells separated from spleen of wild-type C57BL/6 mice using T-cell antibody in anti-CD3- and anti-CD28-coated 96-well plates at different MDSCs: T cells ratios. T cells were labeled with 5 μM carboxyfluorescein succinimidyl ester (CFSE) in complete RPMI and the pro-liferation was evaluated by flow cytometry 72 h later.

**RNA-seq analysis and Real-Time quantitative PCR (RT-qPCR)**. Total RNA of cancer cells or tumor tissues was isolated using TRIzol Reagent (#15596026, Invitrogen) according to the manufacturer's instructions. Then the RNA samples were submitted to the Novogene Genomics Corporation and were sequenced on Illumina NovaSeq 6000 platform. Raw reads were mapped to the human reference genome (hg38) or mouse reference genome (mm10). The analysis results with cutoff (p-value<0.05, fold change>1.5) were used in the analysis with Cluster 3.0 and DAVID 6.8 (https://david.ncifcrf.gov/tools.jsp). Two biological replicates and our published dataset (GEO Accession Number: GSE106306) for human samples[24], and three for mouse samples were used for analysis. The raw data are available on www.ncbi.nlm.nih.gov/geo/query/acc.cgi?acc=GSE166513.

For RT-qPCR analysis, each RNA sample was reverse transcribed with the RevertAid First Strand cDNA Synthesis Kit (#K1621, Thermo Fisher Scientific) using Oligo (dT) primers. Relative quantitation was determined using the LightCycler 480 real-time PCR system (Roche) and then calculated by means of the comparative Ct method (2$^{-\Delta\Delta Ct}$) with the expression of GAPDH as control. Each sample was examined at least in triplicate. The primers used were listed in Supplementary Table 2.

**Western blot**. Cells or fresh tissues were lysed on ice using RIPA lysis buffer (Solarbio, Beijing, China) with freshly added proteinase inhibitors cocktail (Roche). The protein extracts were clarified and quantified by Pierce BCA Protein Assay (Thermo Scientific). Western blot was performed according to our published procedure[24]. The primary antibodies used as follows: anti-TBX3 (#42-4800, Invitrogen, 1:50), anti-Tubulin (#T5168, Simga, 1:10000), anti-BRAF (#14814, CST, 1:2000), anti-c-Jun (#9165, CST, 1:2000), anti-ERK1/2 (#4695, CST, 1:2000), anti-p-ERK1/2 (#4370, CST, 1:2000), anti-JunB (#3753, CST, 1:1000), anti-c-Fos (#2250, CST, 1:1000), anti-GAPDH (#5174, CST, 1:1000), anti-β-Actin (#3700, CST, 1:1000), anti-p65 (#8242, CST, 1:1000), anti-p-p65 (#3033, CST, 1:1000), anti-IKKβ (#8943, CST, 1:1000), anti-p-IKKα/β (#2697, CST, 1:1000), TLR2 (#bs-1019R, Bioss, 1:1000), anti-AKT (#4691, CST, 1:2000), anti-p-AKT (#4060, CST, 1:1000), anti-p38 (#8690, CST, 1:1000), anti-p-p38 (#4511, CST, 1:1000), anti-JNK (#9252, CST, 1:1000), anti-p-JNK (#9255, CST, 1:1000), anti-STAT3 (#9139, CST, 1:1000), anti-p-STAT3 (#9145, CST, 1:1000).

**In situ hybridization**. Tumors from mPTC or mPTC-TAM were frozen in optimal cutting temperature compound (SAKURA) at −80 °C, and sectioned at a thickness of 10 μm. TBX3 in situ hybridizations was performed according to previous protocol[60]. Digoxigenin-UTP labeling and in vitro transcription of plasmid cDNA into antisense were performed using the DIG RNA Labeling Kit (Roche) following the manufacturer's instructions. Hybridization was performed in 150 μl hybridi-zation solution (10 mM Tris-HCl [pH = 7.5], 600 mM NaCl, 1 mM EDTA [pH = 7.5], 0.25% SDS, 10% Dextran Sulfate, 1 × Denhardts [Thermo], 200 μg/ml yeast tRNA [Thermo], 50% formamide) with TBX3 probe per slide at 65 °C overnight. Slides were blocked in 20% sheep serum and 2% BMB for 1 h and then incubated in anti-Digoxigenin-AP antibody (Roche, 1:2500) with 5% sheep serum and 2% BMB at 4 °C overnight. In situ signals were detected using BM purple substrate (Roche).

**Antibody array**. The profiles of chemokines were examined using a Human Cytokine Antibody Array G3 (#AAH-CYT-G3-4, RayBiotech). The supernatant obtained from K1 or TPC1 cells with TBX3 knock-down was analyzed according to the manufacturer's instructions by the Guangzhou RayBiotech Corporation, China. The levels of certain chemokines were listed in Supplementary Data 2.

**ELISA analysis**. Mouse blood was obtained and centrifuged at 1000 g at 4 °C for 10 min after clotting for 30 min on ice to harvest the serum, which was further centrifuged at 12,000 g at 4 °C for 10 min to separate residue. Samples were sub-jected to ELISA analysis. Serum T3, T4, TSH, CXCL1 and CXCL2 levels were determined by mouse ELISA kits (Genie). Human CXCL1, CXCL2, and CXCL8 protein levels in culture supernatants of K1 and TPC1 cells with knock-down of TBX3 were measured using human ELISA Kits (Abcam). All experiments were performed according to the manufacturer's protocols.

**Immunohistochemistry and immunofluorescence analysis**. Tissue specimens or tumor issues from mouse models were fixed in 4% PFA and embedded in paraffin. Immunohistochemistry and immunofluorescence were performed following a standard procedure[5]. 5 μm sections were stained with hematoxylin and eosin (H&E) and incubated with primary antibodies including: anti-Ki67 (#ab16667, Abcam, 1:200), anti-CXCL1 (#GTX31184, GeneTex, 1:2000), anti-CXCL2 (#GTX31171, GeneTex, 1:1000), anti-CXCL8 (#T5153, ImmunoWay, 1:500), anti-CD11b (#ab133357, Abcam, 1:1000), anti-Gr-1 (#108401, Biolegend, 1:50), anti-MRP8 (#ab92331, Abcam, 1:2000), anti-CD4 (#ab183685, Abcam, 1:500), anti-CD8 (#98941, CST, 1:200), anti-TBX3 (#16741-1-AP, Proteintech, 1:200), anti-Thyroglobulin (#ab156008, Abcam, 1:400), anti-CXCR2 (#ab225732, Abcam, 1:100), anti-Gr-1 (#550291, BD, 1:100). IHC slides were scanned with a light microscope. The immunoreactive score (IRS Score) was calculated by multiplying percentage of positive cells and immunostaining intensity[24]. The final score yielded a range from 0 to 12: 0-1 was considered to be no expression, 2–4 was low expression, 5–8 was moderate expression, while >9 was high expression. Tissue specimens of normal tissues, pathological grade I, II, III, and IV were grouped into Normal, Grade low (I and II) and Grade high (III and IV) for a few of grade II according to the pathological criteria of PTC.

**Luciferase reporter construct and reporter activity assay**. To generate reporter constructs for luciferase assays, segments containing AP-1 predicted binding sites at -149 bp, -358bp, -362bp and -1295bp upstream of the promoter according to Jasper database (http://jasper.genereg.net) with the stringency of 80% were cloned into the GLuc reporter plasmid (Gene Copoeia., Guangzhou, China). Additionally, -149 site: 5′-ATAACTCA-3′ was mutated to 5′-GCGGTGAC-3′, and cloned into Gluc plasmid. HEK293T cells were seeded in 24-well plate and co-transfected with AP-1 expression plasmids and different promoter reporter constructs. The luci-ferase activities were measured with the Secrete-PairTM Dual Luminescence Assay Kit (Gene Copoeia, Guangzhou China) according to the manufacturer's protocol 48 h later. For luciferase activity of TLR2 regulated by TBX3, full length (2200 bp) for TLR2 promoter was cloned into GLuc plasmid and then co-transfected with TBX3 or TBX3$^{\Delta A}$ (deletion of activator domain) into HEK293T for further measurement.

**Chromatin immunoprecipitation (ChIP)**. Approximately 5 × 10$^7$cells were used for each ChIP assay according to a previous study[61]. Chromatin was sonicated using a Bio-Red sonicator for several rounds on ice until the de-crosslinked DNA fragment was about 200-500 bp. Flag beads was then incubated overnight at 4 °C with the RIPA150 (50 mM Tris-HCl [pH = 8.0], 150 mM NaCl, 1 mM EDTA [pH = 8.0], 0.1% SDS, 0.5% Triton X-100). Immune complexes were then washed twice with RIPA150 buffer, twice with RIPA500 and once with TE buffer (10 mM Tris-HCl [pH = 8.0], 1 mM EDTA [pH = 8.0]). Then purified DNA was subject to qPCR assays with specific primers. The primers used were listed in Supplementary Table 2.

**DNA affinity immunoblot assay**. Biotinylated DNA probe was incubated with Dynabeads Streptavidin M-280 (#11205D, Invitrogen) according to the manu-facturer's instructions. The M-280-probe was then incubated with K1 nuclear extracts in binding buffer (20 mM Tris-HCl [pH = 8.0], 150 mM NaCl, 1 mM EDTA [pH = 8.0], 5% glycerol, 1 mM DTT, 1 mM PMSF) at 4 °C for 6 h[24]. Binding samples were washed with washing buffer (10 mM Tris-HCl [pH = 8.0], 200 mM NaCl, 1 mM EDTA [pH = 8.0], 1 mM DTT, 0.1% NP40) for five times and boiled in protein loading buffer. Western blot was performed to analyze the AP-1 complex levels. Biotinylated DNA oligos were designed as follows: Control: 5′-biotin-ATAATATG ACTAAGTACAATTAACCGCTAGATCAGACGATTAGCCGTGTCCACTATGG A-3′, 149-WT: 5′-biotin-GTTACAGTTTCCGACACCTTCTTTTTATAACTCAGCT CTATCCCCCAGCACTCGACCTG-3′, 149-MUT: 5′-biotin-CGTTACAGTTTCCG ACACCTTCTTTTTGCGGTGACGCTCTATCCCCCAGCACTCGACCT-3′.

**Flow cytometry**. Mouse tumors were minced into small fragments on ice and then separated into single cells with tissue dissociation buffer including 1 mg/ml col-lagenase I (Sigma) and 0.5 mg/ml dispase II (Invitrogen) dissolved in PBS at 37 °C for 60-90 min. Samples were mashed through 40μm filters into FACS buffer (1% FBS in PBS) and washed for three times. The single cells were stained with Zombie NIR™ Fixable Viability Kit (#423105, Biolegend, 1:1000) for 15 min in dark on ice, and then incubated with anti-CD16/CD32 (#553141, Becton Dickinson, 1:100) for blocking and other indicated antibodies for 30 min on ice. Antibodies were showed as follows: AF-700 anti-CD45 (#147716), FITC anti-CD45(#103108), PE anti-F4/80 (#123110), Pacific Blue anti-CD11b (#101224), FITC anti-CD3 (#100306), Percp-cy5.5 anti-CD3 (#100328), APC anti-NK1.1 (#108709), BV605 anti-CD4 (#100451), AF-700 anti-CD4 (#100430), BV510 anti-CD8 (#100752), Percp-cy5.5 anti-Gr-1 (#108428), BV605 anti-Ly-6G (#127639), BV510 anti-Ly-6C (#128033), PE-CF594 anti-CD11c (# 117347), PE-Cy7 anti-CD19 (#115519), PE-Cy7 anti-Ly-6C (# 128017), APC anti-CD115 (# 135509), PE anti-CD34 (# 152203), PE-CF594 anti-CD135 (# 313319), BV421 anti-CD117 (# 105827), FITC anti-CD16/32 (# 101305), AF-700 anti-Sca-1 (# 108141), Percp-cy5.5 anti-Ter-119 (# 116227), APC anti-IFN-γ (# 505809), PE anti- Granzyme B (#396405) at dilution 1:200 were all purchased from Biolegend. Flow cytometry was performed using standard protocol

on LSRFortessa analyzer (Becton Dickinson) and analyzed with FlowJo V10 software.

**Statistical analysis**. All data are expressed as the mean ± s.d. and the significance of difference of two groups was determined by two-tailed Student's $t$ test. Statistical analysis was performed using SPSS 17.0 software (IBM Corporation, Armonk, NY) and GraphPad Prism 8.0 software (La Jolla, CA, USC). $P$ value < 0.05 was considered statistically significant. Datasets (GEO Accession Number: GSE126153 and GSE126154) were performed to analyze TBX3-regulated genes and TBX3 binding sites at the promoter region of TLR2. Datasets (GSE161430, GSE75299, GSE152699) with cutoff of fold change > 1.5 and $P$ value <0.05 were used to analyze MAPKi-repressed genes.

**Reporting summary**. Further information on research design is available in the Nature Research Reporting Summary linked to this article.

## Data availability

The RNA-seq data generated in this study have been deposited in the Gene Expression Omnibus (GEO) under the accession GSE166513 available from. GEO datasets (GEO Accession Number: GSE126153, GSE126154, GSE161430, GSE75299 and GSE152699) and TCGA-THCA dataset [https://portal.gdc.cancer.gov/projects/TCGA-THCA] are used for analyses in this study. All remaining data associated with this study are available within the Article, Supplementary Information or Source Data file. Source data are provided with this paper.

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

## Acknowledgements

The authors would like to thank Dr Anne Moon for kindly providing the *Tbx3^{flox/flox}*, *Tbx3^{GFP/+}* genetics mouse models. We also thank Dr Martin McMahon for kindly sharing the *Braf^{CA}* (*LSL-BRAF^{V600E}*) mouse model, Dr Shioko Kimura for sharing the *TPO-Cre* mouse model. We thank Dr Long Li from Tianjin Medical University for project design and FACS analysis of immune cells. This work was supported in part by the National Natural Science Foundation of China (81872235 and 82073052 to L.Z., 81702710 to S.Y., 81971650 to Z.M., 81872169 to X.Z.); the Postdoctor Fund of China (2020M680897 to P.Z.) and the Tianjin Research Innovation Project for Postgraduate Students (2020YJSS172 to Y.L.).

## Author contributions

L.Z. designed the project; P.Z., H.G., S.Y., J.Z., Z.Z., and Y.L. performed the experiments and analyzed the results. X.Z., Z.M., and Y.Y. provided the clinical ideas, suggestions, and samples. L.Z. and P.Z. wrote the manuscript.

## Competing interests

The authors declare no competing interests.
