## [Peer Review File · Nature Communications]

Reviewers' Comments:

Reviewer #1:

Remarks to the Author:

The manuscript titled "Targeting a BRAFV600E-TBX3-MDSCs Axis Reverts Immune Suppression and Sensitizes Thyroid Carcinoma to MKIs Therapy" by Zhang, Guan, Yuan, et al. describes an important signalling axis that promotes papillary thyroid cancer (PTC) development and whose disruption has therapeutic potential. Through a series of in vitro and in vivo experiments the authors show that Tbx3 is transcriptionally upregulated by the AP-1 proteins, c-Jun, JunB and c-Fos that function downstream of BRAFV600E and that Tbx3 is required for BRAFV600E induced PTC tumours. The manuscript provides novel evidence that TRL2 is transcriptionally activated by TBX3 which leads to upregulated expression of the CXCR2 ligands in a NFkB-dependent manner followed by myeloid-derived suppressor cell (MDSCs) infiltration which contribute to tumour formation. In my opinion, this manuscript represents an impressive body of novel work that follows a logical experimental flow and that it is worthy of publication in your journal provided that the authors address the following comments:

General comments

1. Densitometric analyses for all western blotting are needed and the molecular weights of the proteins shown should be added.
2. To facilitate the reading of the manuscript and to appreciate the data more information on names of inhibitors and genes expressed should be included on the figures. For example, in Figure 2d, the AP-1 inhibitor SR11302, Figure 4e, "Relative TRL2 mRNA levels".

Introduction:

1. The heading "Introduction" is missing.
2. Page 3 sentence on line 45-47: Requires a reference.
3. Line 60: More information on "PLX4032" should be provided.
4. The authors should describe the known link between BRAF and TBX3 in melanoma.

Materials and Methods:

1. The °C sign is displayed incorrectly.
2. ATCC reference numbers for all cell lines used should be provided.

Results:

Figure 1:

1. Figure 1c: Why are there two bands for TBX3? Does it possibly represent the two TBX3 isoforms?
2. Figure 1d would benefit by quantification of the tumour sizes for the respective mouse strains in a similar manner to figure 1k.
3. Figure 1e, is missing the images for the mPTC+/- tumors. How do the authors explain the disrupted follicle architecture observed in the mPTC+/- tumors?
4. Figure 1f and g: Why include lines between the different groups?
5. In figure 1k, images should be provided for the WT mice.
6. Information in the Figure legend for 1j and 1k are swapped.

Figure 2:

1. Figure 2c: the names of the BRAF and ERK1/2 inhibitors should be included in the text. Line 159, the effect of the BRAF inhibitor was not dose-dependent and this should be reworded.
2. Figure 2d: the name of the AP-1 inhibitor SR11302 should be added to the figure and evidence should be provided that it does indeed inhibit AP-1 proteins.
3. Figures 2h: Why is the effect of overexpressing c-Jun, JunB and c-Fos individually the same as when they co-expressed?
4. Figure 2m: The authors should include the location of the primers used for the ChIP assay on Figure 2k. What internal control was used? Furthermore, why was FLAG-AP-1 used instead of endogenous AP-1?
5. Figure 2n: What BRAF and ERK1/2 inhibitors were used? This information should be added in the figure and legend.
6. Figure 2o: the p-value is indicated as being 0 which can't be. The authors need to correct this.
7. To confirm that the AP-1 site at -149 is responsible for AP-1 binding and for mediating the

effects of AP-1 on the TBX3 promoter, it should be mutated and compared with WT in luciferase reporter and in vitro DNA binding assays.

Figure 3:

1. Figure 3c: the authors should describe the results for MMP9, Klf9, Met and Six4.

Figure 4:

1. Figure 4a: the inhibitors are represented in the following order: IKKB, CREB, AP-1 and STAT3 whereas the text describing the figure has a different order, the authors should amend this.

2. Figures 4a and b: the labels below the graphs "+vector" and "+TBX3" are not correctly aligned.

3. Line 226: the figures referred to should include Supplementary figure 4d.

4. Line 246: the description of the data should mention the transcriptional activation and then the binding of TBX3.

Figure 5:

1. Figure 5b: The following statement "Cells showed different sensitivity to chemokine doses, but CXCL1/2 double loss and CXCL8 single loss exhibited similar effects" is not true for K1 cells.

2. Line 263: should read "CXCR2 inhibitors".

3. Line 274: fig 7g should be replaced with fig 5g.

Figure 6:

1. Figures 6c and 7d: Which MDSC subpopulation was used?

Figure 7:

1. More information on Gr-1 should be provided.

2. The figure legend should indicate what mPTC -1m, 2m, 3m, 6m and 12m mean.

Supplementary Figures:

1. Figure legend for supplementary figure 1a: the information for H is incorrect.

2. Supplementary Figure 1h: the image refers to mPTC/Tbx3G/+but the legend says mPTC/Tbx3G/-.

3. Supplementary Figure 4d: the labels below the graphs "+vector" and "+TBX3" are not correctly aligned.

Line 297: "normal" should be removed from "normal PTC cells".

Prof Sharon Prince

Reviewer #2:

Remarks to the Author:

Zhang P et al describe the role of TBX3 in BRAFV600E-driven papillary thyroid cancer progression. They find that the BRAF/MAPK pathway activation induces TBX3 expression through AP-1 mediated transcriptional regulation. Moreover, they identify CXCR2 ligands Cxcl1, 2 and 8 expression to be increased by TBX3 and mediating tumor cell growth.

The identified mechanism is of great relevance for the understanding of BRAFV600E-driven PTC and the experiments in the different mouse and human model systems are very well designed. The results throughout are convincing.

However, the manuscript should be revised for English language. While the data is convincing, it would help if the figures would be annotated in more detail to be more self-explanatory. A strength of the paper is the consistent results in different models of thyroid cancer. However, the genotype of the human cell lines (K1, TPC1 and BCPAP) should be stated.

Specific Comments:

1.) Abstract and Title:

- The authors use the term multikinase inhibitors too loosely, which can be misleading. In their manuscript they study the effect of targeted BRAF-inhibition with vemurafenib, which is relatively selective for RAF kinases. The term multikinase inhibitors is better applied for compounds that target multiple kinases (i.e. VEGFR and others).

2.) Introduction

- Page 3, line 45-47: As above. The authors should distinguish between multikinase inhibitors such as Sorafenib and Lenvatinib, which have shown activity against radioiodine refractory thyroid cancer (Brose MS et al, Lancet 2014; Schlumberger M et al, NEJM, 2015) and the combined

BRAF/MEK inhibition in BRAFV600E mutant PTCs, which has more modest activity and has not been approved by the FDA for this indication.

3.) Results

Figure 1:

- Heading: Stating that Tbx3 is necessary for BrafV600E to induce PTC seems an overstatement based on the data presented.
- Figure 1f and g: the results are difficult to read. The interconnecting lines between genotypes are not explained. Is this intended to show difference between littermates? It would help, if the authors would represent the thyroid weights and mouse weights similar in figure 1k, or in boxplots.
- In the IHC images in 1e and 1j the differences between the different mouse models are challenging to identify. Would the authors be able to either provide better quality pictures (perhaps with an inset at higher magnification) and/or quantify the Ki67-staining?

Figure 2:

- Figure 2a: Could the authors provide the rationale for overexpressing BRAF in the BRAFV600E mutant K1 cell line? This line is more appropriate for the KD experiment shown in 1b. Moreover, the genetic driver of the TPC1 cell line is known to be a RET fusion, meaning that MAPK pathway is constitutively activated but subject to ERK feedback. What is the rationale for creating a BRAF overexpressing TPC1 cell line?
- Figure 2l-m) please add for clarification the model system/cell line to the respective graph.

Figure 3:

- Figure 3b: Please add a title to the graphs showing what is compared to what. Please provide the list of the 191 genes overlapping between mouse and human.
- Figure 3e: consider breaking up the Y-axis to better display the differences between shCtrl and the two TBX3 short hairpins for all genes. The increase in Pdgfb is intriguing. Was there any difference in endothelial or stromal cell elements in the Tbx3-null tumors?
- Figure 3f: The differences between shctrl and shTBX3 seem to be very subtle and they are not well shown in the image provided.

Figure 4:

- Figure 4a and b: both figures are very hard to interpret, since it is not clearly visible what is compared to what.
- Figure 4d: Why is breast cancer and thyroid cancer compared at this point? Please provide the list of the overlapping 76 genes.

Figure 6: The authors describe a reduction in 'G-MDSC' in the context of TBX3^{-/-} mPTCs. However, the authors define MDSCs in general as cells expressing CD45+CD11b+Gr1+. Cells expressing the aforementioned markers can be a wide variety of cells in the tumor microenvironment such as Monocytes, Neutrophils, G-MDSC, M-MDSC, immature Macrophages etc. While it is interesting that the authors find a difference in abundance in the context of TBX3^{-/-}, based on their data the assumption that this population represents MDSCs cannot be made without additional characterization. Since the authors have performed bulk RNA seq on the whole mouse tissue, they could perform immune deconvolution to identify the presence of different immune populations.

Figure 8: Zhong et al show rRNA and protein expression of sorted tumor cells from mouse PTC w/o TBX3^{-/-}. These are important results, that should be included earlier in figure 1 or 2. Moreover, since the hypothesis is that the mechanism by which TBX3^{-/-} mediates progression is cell autonomous, would it be possible to compare RNA seq from sorted tumor cells with human K1 cells instead of whole mouse tissue with K1 cell line as performed in figure 3b?

Supplementary figures:

Suppl. Figure 1:

- 1a) What about the DEGs in mouse and human PTCs or cell lines, which would be more relevant to this study than gene expression changes in melanomas? See Montero-Conde C. Cancer Discov 2013 for human thyroid cancer cell lines; Dunn L. JCEM 2018 for patient biopsy specimens; Saqcena M Cancer Discov 2021, for GEMM of Braf PTCs.

- 1e) In the y axis: relative Tbx3 mRNA
- 1e-g): Please add comparison to WT thyroid

Reviewer #4:

Remarks to the Author:

Manuscript "Targeting a BrafV600E-TBX3-MDSC Axis Reverts Immune Suppression and Sensitizes Thyroid Carcinoma to MKIs Therapy" by Zhang et al, aims to investigate: 1) the role of developmental factor TBX3 in mediating the oncogenic functions of BrafV600E mutation in thyroid cancer and 2) the immunoregulatory effects of TBX3-directed TLR2-NFKB-CXCR2 ligand signaling on the recruitment of MDSCs to tumors. The manuscript utilizes multiple in vivo models, clinical data and in vitro platforms to substantiate the primary role of TBX3, and its down-stream targets TLR2-NFKB-CXCR2, in the initiation and progression of BrafV600E-induced papillary thyroid cancer (PTC) and in the infiltration of MDSCs to PTC tumors. The induction of TBX3 in thyroid cancer cells was the result of the activation of the BrafV600E-MAPK-AP1 axis. Overall, this manuscript is timely and provides significant insight into the previously underexplored role of TBX3 in thyroid cancer and contributes to the field's understanding of immunoregulatory axes in BrafV600E tumors. Also, results could serve as the foundation for new therapies for PTC. Noticed limitations are as follows:

Major comments/suggestions:

- 1) Discrimination of the effects of tumor cell related CXCR2 vs. stroma-myeloid cells expressed CXCR2 in the growth of BrafV600E-induced PTC in mice remains unknown. Although investigators show results demonstrating the crucial role of tumor cell related CXCR2 in the growth of PTC tumors, evaluating the interaction of tumor produced CXCL1, CXCL2 and CXCL8 with the CXCR2 in the stroma would be highly complementary.
- 2) It remains unclear how the upregulation of CXCR2 in BrafV600E-TBX3-induced papillary thyroid tumors promoted the infiltration of PMN-MDSCs. Is CXCR2 signaling in tumor cells promoting the expression of factors that recruit PMN-MDSCs? Is the TBX3 driven production of CXCL1, CXCL2 and CXCL8 attracting CXCR2+ PMN-MDSCs? This should be tested experimentally.
- 3) Authors refer to MDSCs without developing functional assays of T cell suppression. It is important to validate the suppressive function of MDSCs and determine if TBX3 deletion modulates only the accumulation or also the direct T cell suppressive potential of PMN-MDSCs.
- 4) The impact of thyroid cancer cell TBX3 in the immunosuppressive myelopoiesis occurring systemic remains unclear. Establish whether TBX3 expression impacts development of MDSCs from bone marrow precursors or if its effects are predominantly via recruitment is crucial.
- 5) Elucidating whether elimination of T cells (or development of experiments in T cell deficient mice) overcomes the anti-tumor effect induced by TBX3 deletion in BrafV600E-induced PTC will confirm the key role of T cells in the reported anti-tumor responses and the interaction of tumor cell TBX3 and T cell dysfunction. This should be complemented with expression of cytotoxic mediators on T cells.
- 6) It remains unknown whether the higher numbers of T cells present in TBX3-null tumors carry higher expression of checkpoint mediators. This could open the possibility of combination strategies with checkpoint inhibitors.

Minor comments/suggestions:

- 1) Densitometric quantification of western blots is suggested.
- 2) Clarify differences in tumor growth in Figure 5f for CXCL1/2 and CXCL8 overexpression models- Why does OE of CXCL2 or CXCL1 provoke the same effects while CXCL8 OE does not
>Do CXCL1/2 and CXCL8 exhibit differential affinity for CXCR2?
>Do CXCL1/2 and CXCL8 activate the same downstream effects at the same magnitude?
- 3) Include the empty vector + CXCL1/2/8 tumor growth curves/weights in the same figures alongside the shTBX3 OE data (Figure 5f)

Re: Manuscript NCOMMS-21-20886

We have undertaken revisions in response to the reviewers' comments and would now like to submit a revised manuscript. The detailed responses and revisions are given below:

Reviewer #1 (Remarks to the Author): with expertise in Tbx3 biology

The manuscript titled "Targeting a BRAF^{V600E}-TBX3-MDSCs Axis Reverts Immune Suppression and Sensitizes Thyroid Carcinoma to MKIs Therapy" by Zhang, Guan, Yuan, et al. describes an important signaling axis that promotes papillary thyroid cancer (PTC) development and whose disruption has therapeutic potential. Through a series of in vitro and in vivo experiments the authors show that Tbx3 is transcriptionally upregulated by the AP-1 proteins, c-Jun, JunB and c-Fos that function downstream of BRAF^{V600E} and that Tbx3 is required for BRAF^{V600E} induced PTC tumours. The manuscript provides novel evidence that TRL2 is transcriptionally activated by TBX3 which leads to upregulated expression of the CXCR2 ligands in a NFkB-dependent manner followed by myeloid-derived suppressor cell (MDSCs) infiltration which contribute to tumour formation. In my opinion, this manuscript represents an impressive body of novel work that follows a logical experimental flow and that it is worthy of publication in your journal provided that the authors address the following comments:

General comments

1. Densitometric analyses for all western blotting are needed and the molecular weights of the proteins shown should be added.

Response: Thank the reviewer for the suggestion. We marked the molecular weights of target bands into each western blot panel. We also performed

densitometric analysis and presented all related panels in Supplementary Fig. 10-12.

2. To facilitate the reading of the manuscript and to appreciate the data more information on names of inhibitors and genes expressed should be included on the figures. For example, in Figure 2d, the AP-1 inhibitor SR11302, Figure 4e, “Relative TRL2 mRNA levels”.

Response: Thank the reviewer very much for pointing out the unclear statements. We went through all the figures and made the corrections. Related revisions were included in the figures and figure legends.

Related revision was made on Figure 2d, 2n, 4e and Supplementary Fig. 4h.

Introduction:

1. The heading “Introduction” is missing.

Response: We have added the heading “Introduction” to the manuscript.

2. Page 3 sentence on line 45-47: Requires a reference.

Response: Thank the reviewer very much. We have added the reference according to the manuscript.

Manuscript revision was made on page 3 line 45-48: “Nevertheless, targeted therapies with MAPK inhibitor drugs (MAPKi), including Vemurafenib or Dabrafenib, exhibited reserved efficacy on BRAF^{V600E}-positive metastatic, RAI-refractory DTC patients due to primary or acquired drug resistance⁶.”

3. Line 60: More information on “PLX4032” should be provided.

Response: Thank you so much for pointing out the incomplete description. PLX4032, as a BRAF^{V600E} specific inhibitor, selectively blocks the BRAF/MAPK pathway in BRAF mutant cells, typically including thyroid cancer and melanoma cells. Thus, we updated the related statements in the manuscript.

Manuscript revision was made on page 3 line 59-62: “Our previous study found that combined therapy with BRAF^{V600E} inhibitor PLX4032 and anti-PD-L1

antibody inhibited PTC development more efficiently than either single treatment, partially through restoring tsMHCII level.”

4. The authors should describe the known link between BRAF and TBX3 in melanoma.

Response: Thank the reviewer for bringing up this question. In melanoma, constitutive activation of MAPK pathway caused by BRAF^{V600E} mutation up-regulates TBX3, and promotes tumor cell invasion and migration. We have added the description into the introduction.

Manuscript revision was made on page 4 line 84-86: “Interestingly, TBX3 was predominantly involved in BRAFV600E-induced MAPK pathway activation and melanoma invasion, which proposes the potential regulation of BRAF^{V600E} on TBX3 through other types of tumorigenesis²⁶.”

Materials and Methods:

1. The °C sign is displayed incorrectly.

Response: Sincerely appreciate the careful review of the manuscript. We have corrected the related format of “°C”, and related revisions were made in several places.

2. ATCC reference numbers for all cell lines used should be provided.

Response: We apologize for our careless presentation. Actually, HEK293T cells were obtained from ATCC with *Num.:* ACS-4500. As for cancer cell lines, they were all gifted by Professor Ming Gao from Tianjin Medical University Cancer Institute and Hospital with related STR profiling. The same batch of cells as those used in our previous study¹.

Manuscript revision was made on page 21 line 519-521: “HEK293T (ACS-4500) was purchased from the American Type Culture Collection (ATCC). And all cancer cell lines were obtained from Tianjin Medical University Cancer Institute and Hospital with STR profiling,”

Results:

Figure 1:

1. Figure 1c: Why are there two bands for TBX3? Does it possibly represent the two TBX3 isoforms?

Response: We have checked our western blot with another TBX3 primary antibody (ABE807; Millipore) and confirmed that the band with arrowhead was the TBX3 band, while the other bands were background. As the two isoforms of TBX3 are concerned, the difference is only 20 amino acids that are hard to be detected by western blot with this precision.

Related revision was made on Fig. 1c. Arrowheads were added into the western blot result to mark the main TBX3 bands.

2. Figure 1d would benefit by quantification of the tumour sizes for the respective mouse strains in a similar manner to figure 1k.

Response: Thank the reviewer very much for your suggestion. We quantified the tumor sizes from Fig. 1d in the similar manner to Fig.1k and combined Fig.1f into Fig.1d.

Related revision was made on Fig. 1d. We combined the quantification of tumor sizes into Fig.1d.

Manuscript revision was made on page 6 line 130 to 131: “Quantification further confirmed that Tbx3 loss inhibited initiation and development of BRAF^{V600E}-induced PTC in a dose-dependent manner (Fig. 1d).”

3. Figure 1e, is missing the images for the mPTC+/- tumors. How do the authors explain the disrupted follicle architecture observed in the mPTC+/- tumors?

Response: Thank the reviewer very much for your suggestion. We added the mPTC/Tbx3^{+/-} images to Fig. 1e, which helps us to better understand the dose-dependent function of TBX3 in mPTC development. Just because haploinsufficiency of TBX3 also impeded mPTC occurrence, we observed disrupted follicle architecture in mPTC/Tbx3^{+/-} tumor areas.

Related revision was made on Fig. 1e. We added the mPTC/Tbx3^{+/-} images to Fig. 1e

4. Figure 1f and g: Why include lines between the different groups?

Response: Thank the reviewer very much for the question. To better demonstrate that the dose of TBX3 has an important effect on tumor size and body weight, we conducted the comparison among litter-mates and each line represents one litter. Actually, we also found that BRAF^{V600E}-induced mPTC dosely depends on TBX3 by roughly comparing tumor weight and body weight across three genotypes.

Related revision was made on Fig. 1d and Fig.1f. We performed the tumor weight and body weight comparison across three genotypes and summarized in Fig.1d and Fig. 1f.

5. In figure 1k, images should be provided for the WT mice.

Response: According to your suggestion, we added representative picture of WT and mPTC-CON mice thyroids into Supplementary Fig. 1j. The volume of mPTC-CON thyroid is slightly bigger than WT, which may be due to CreER activity leak².

Related revision was made on Supplementary Fig. 1j. We added representative picture of WT and mPTC-CON mice thyroids into Supplementary Fig. 1j.

6. Information in the Figure legend for 1j and 1k are swopped.

Response: Thank the reviewer for kindly pointing out our mistake, we have corrected the related statements in the figure legends.

Related revision was made on Figure legend for Fig. 1i and 1j.

Figure 2:

1. Figure 2c: the names of the BRAF and ERK1/2 inhibitors should be included in the text. Line 159, the effect of the BRAF inhibitor was not dose-dependent and this should be reworded.

Response: Thank the reviewer for pointing out the inappropriate description, BRAF inhibitor PLX4032 and ERK1/2 inhibitor SCH772984 were used in our experiments. Actually, we are pretty sure the effect of the inhibitors was dose-dependent based on our multiple rounds of experiments. We apologized that the most representative result was not included in Fig. 2c, which has been replaced with newly repeated result.

Related revision was made on Fig. 2c. Newly repeated result was incorporated into Fig. 2c.

2. Figure 2d: the name of the AP-1 inhibitor SR11302 should be added to the figure and evidence should be provided that it does indeed inhibit AP-1 proteins.

Response: Thank the reviewer for the suggestion. The efficiency of SR11302 on AP-1 inhibition was confirmed with the reduction of AP-1 target gene expression, without change in AP-1 expression itself according to previous study^{3,4}. Our experiments demonstrated that TBX3 transcription, in addition to the well-recognized AP-1 target gene c-Myc, were both repressed upon SR11302 application in K1 cells. These results confirmed both the inhibitory effect of SR11302 on AP-1 function and the regulation of AP-1 on TBX3 transcription.

Related revision was made on Fig. 2d and related reference was added.

3. Figures 2h: Why is the effect of overexpressing c-Jun, JunB and c-Fos individually the same as when they co-expressed?

Response: We indeed thought about the question carefully. Our guess is that overexpression of any AP-1 protein is enough to occupy the related binding site on *TBX3* promoter region since they share the same binding pattern, which has been confirmed by our ChIP and promoter GLuc assays. As we mentioned in the discussion, different AP-1 sites were employed in breast cancer development and PTC, which could relate very much to the epigenetic environment.

4. Figure 2m: The authors should include the location of the primers used for the ChIP assay on Figure 2k. What internal control was used? Furthermore, why was FLAG-AP-1 used instead of endogenous AP-1?

Response: Thanks for your suggestion. Based on our truncation and mutation analysis, we think the main binding site was -149 site. According to your suggestion, ChIP experiments with antibodies against endogenous AP-1 proteins were conducted to further confirm the direct binding site. The primers used for the ChIP assay were included in Fig. 2k, and primers targeting

GAPDH gene were used as the internal control⁵.

Related revision was made on Figure 2k, 2n and the legends. The primers used for ChIP assay were included in Fig. 2k. ChIP experiments with antibodies against endogenous AP-1 proteins were conducted and replaced with Fig. 2n.

Manuscript revision was made on page 8 line 182-183: “Specific mutation and endogenous ChIP experiments also showed that AP-1 proteins were recruited to -149bp site (Fig. 2m, n).”

5. Figure 2n: What BRAF and ERK1/2 inhibitors were used? This information should be added in the figure and legend.

Response: We supplemented Fig. 2n with PLX4032 (BRAF inhibitor) and SCH772984 (ERK1/2 inhibitor) both in the figure and legend.

Related revision was made on Fig. 2o.

6. Figure 2o: the p-value is indicated as being 0 which can't be. The authors need to correct this.

Response: Thank the reviewer for kindly pointing out this question. Due to the extremely low p-value, the softwares provide p-value as “0”. We have corrected it into “p<0.001” in Fig. 2p.

Related revision was made on Fig. 2p.

7. To confirm that the AP-1 site at -149 is responsible for AP-1 binding and for mediating the effects of AP-1 on the TBX3 promoter, it should be mutated and compared with WT in luciferase reporter and in vitro DNA binding assays.

Response: Thank the reviewer for your suggestion. We performed luciferase reporter assay with -149 site mutated construct and in vitro DNA pull down assay, which further confirmed that -149 site mutation repressed AP-1 complex directly bind to TBX3 promoter.

Related revision was made by adding Fig. 2m and Supplementary Fig. 2e.

Mutation and DNA pull down experiment results were included as Figure 2m and Supplementary Fig. 2e respectively.

Manuscript revision was made on page 8 line 182-185: “Specific mutation and endogenous ChIP experiments also showed that AP-1 proteins were recruited to -149bp site (Fig. 2m, n). In addition, we conducted in vitro DNA binding affinity assay, which further confirmed the binding of AP-1 to -149 site (Supplementary Fig. 2e).”

Figure 3:

1. Figure 3c: the authors should describe the results for MMP9, Klf9, Met and Six4.

Response: Thank the reviewer for the suggestion. Previous study demonstrates the oncogenic function of MMP9, MET, and SIX4. MMP9 is involved in the breakdown of extracellular matrix in normal physiological processes and highly expressed in tumor for tumor invasion and migration⁶; MET plays an important role in promoting PTC cell proliferation through PI3K/AKT signaling⁷. SIX4 functions as an oncogene to promote tumor metastasis through upregulating YAP1 and c-MET in hepatocellular carcinoma⁸. While as a proliferation restricting factor, KLF9 is reported to induce Medullary thyroid carcinoma apoptosis and suppress gastric cancer cell invasion and metastasis^{9,10}. We have added relative description and reference into the manuscript.

Manuscript revision was made on page 9 line 206 to 208: “During validation of representative genes by RT-qPCR, we confirmed de-repression of proliferation restricting genes including Cdkn1c, Cdkn2c, and Klf9 upon Tbx3 knock-out, and repression of oncogenic genes such as Igfbp3, Mmp9, Met and Six4.”

Figure 4:

1. Figure 4a: the inhibitors are represented in the following order: IKKB, CREB, AP-1 and STAT3 whereas the text describing the figure has a different order, the authors should amend this.

Response: Thank the reviewer for the suggestion. We amended the related description.

Manuscript revision was made on page 10 line 233 to 236: “In comparison,

repression of p38/CREB, JNK/AP-1, or AKT/STAT3 activity did not show meaningful impact of TBX3-promoted chemokines even though these factors were under TBX3 regulation and have been involved in chemokine regulation (Fig. 4a and Supplementary Fig. 4a-c).”

2. Figures 4a and b: the labels below the graphs “+vector” and “+TBX3” are not correctly aligned.

Response: Thank the reviewer for picking out the mistake. We have corrected Figures 4a and 4b.

Related revision was made on Fig. 4a, 4b and Supplementary Fig. 4a and 4d: To appreciate better, we divided Figure 4a and b into a few independent graphs and place into Supplementary Fig. 4a and 4d.

3. Line 226: the figures referred to should include Supplementary figure 4d.

Response: Thank the reviewer for picking out the mistake, we have corrected the figure referring.

Manuscript revision was made on page 10 line 233 to 236: “In comparison, repression of p38/CREB, JNK/AP-1, or AKT/STAT3 activity did not show meaningful impact of TBX3-promoted chemokines even though these factors were under TBX3 regulation and have been involved in chemokine regulation (Fig. 4a and Supplementary Fig. 4a-c).”

4. Line 246: the description of the data should mention the transcriptional activation and then the binding of TBX3.

Response: Thank the reviewer for pointing out the shortcomings in our description. We have adjusted the logical order in our manuscript.

Manuscript revision was made on page 11 line 256 to 258: “Luciferase activity assay and ChIP experiments showed that TBX3 may transcriptionally activate TLR2 expression by directly binding to the promoter via the activation domain. (Fig. 4h, i)”

Figure 5:

1. Figure 5b: The following statement “Cells showed different sensitivity to

chemokine doses, but CXCL1/2 double loss and CXCL8 single loss exhibited similar effects” is not true for K1 cells.

Response: Thank the reviewer for bringing up this question. To be exact, loss of CXCL1/2 and CXCL8 showed proliferation repression, although CXCL1/2 double loss seems to exhibit stronger effect than CXCL8 loss. The main point we want to deliver is that the chemokines promote cell proliferation downstream of TBX3. As to the slightly different function between CXCL1/2 and CXCL8, it could relate to the binding affinity with CXCR2 receptor.

Manuscript revision was made on page 11 line 268 to 270: “Decreased expression of CXCL1/2 and CXCL8 significantly inhibited PTC cell proliferation, even cells showed different sensitivity to chemokine doses (Supplementary Fig. 5a and Fig. 5a, b)”

2. Line 263: should read “CXCR2 inhibitors”

Response: Thank the reviewer for picking out the mistake. We have corrected our manuscript.

Manuscript revision was made on page 12 line 272 to 274: “we found that CXCR2 inhibitors SCH527123 and Reparixin inhibited PTC cell proliferation in dose-dependent manners even though K1 cells and TPC1 cells responded to different concentrations.”

3. Line 274: fig 7g should be replaced with fig 5g.

Response: Thank the reviewer very much for kindly pointing out the inaccurate statements. We have corrected the related statements in the manuscript.

Manuscript revision was made on page 12 line 281 to 283: “Histologically, CXCL1, CXCL2 and CXCL8 was dramatically reduced in TBX3 deficient tumors simultaneously with Ki67 (Fig. 5f).”

Figure 6:

1. Figures 6c and 7d: Which MDSC subpopulation was used?

Response: We are sorry for the unclear description. G-MDSCs subpopulation was used in Fig. 6c and 7d.

Figure 7:

1. More information on Gr-1 should be provided.

Response: Thank the reviewer for your kind suggestion. From early observations in tumor-bearing mice, MDSCs were characterized by the expression of CD11b and the myeloid differentiation antigen Gr-1, which was generally accepted to analyze the level of MDSC in mice¹¹. Additionally, S100A8 and S100A9 were also used for analyses of MDSCs in tumor-bearing mice^{11,12}. We add a reference about Gr-1 in our manuscript.

2. The figure legend should indicate what mPTC -1m, 2m, 3m, 6m and 12m mean.

Response: Thank the reviewer very much for your kind suggestion. For mPTC -1m, 2m, 3m, the time was for mice age, as for mPTC-TAM-6m, 12m, the time was about tamoxifen-induction time.

Related revision was made on figure legend for Fig. 7g: “FlowJo analysis of G-MDSC, M-MDSC, CD4⁺ T cells and CD8⁺ T cells of CD45⁺ TILs in mPTC-TAM tumors induced with tamoxifen for 6m and 12m (g)”

Supplementary Figures:

1. Figure legend for supplementary figure 1a: the information for H is incorrect.

Response: H represents the overlapped DEGs from GSE152699 and GSE75299, referring to down-regulated genes in human melanoma cells upon MAPKi treatment

Related revision was made on figure legend for Supplementary Fig. 1a:

“H: Overlapped DEGs from GSE152699 and GSE75299, referring to down-regulated genes in human melanoma cells upon MAPKi treatment.”

2. Supplementary Figure 1h: the image refers to mPTC/Tbx3G/+but the legend says mPTC/Tbx3G/-.

Response: Thank the reviewer for pointing out the mistake. We have corrected our figure legend into Tbx3^{G/+}.

Related revision was made on figure legend for Supplementary Fig. 1h:

“(h) The images representing whole thyroid tissues from mPTC and mPTC/Tbx3^{G/+} mice at 5w”

3. Supplementary Figure 4d: the labels below the graphs “+vector” and “+TBX3” are not correctly aligned.

Line 297: "normal" should be removed from “normal PTC cells”.

Response: Thank the reviewer for pointing out the mistake. According to your suggestion, we moved left the “+vector” and “+TBX3”. In addition, it is really a mistake to add normal before PTC cells, and we have removed it. And the original Supplementary Fig 4d was now Supplementary Fig 4c

Related revision was made on Supplementary Fig. 4c.

Manuscript revision was made on page 14 line 325 to 327: “Similarly, conditioned medium from TBX3 knock-down PTC cells showed impaired neutrophil attraction, which was rescued by over-expressed CXCR2 ligands (Supplementary Fig. 7f).”

Prof Sharon Prince

Reviewer #2 (Remarks to the Author): with expertise in thyroid cancer

Zhang P et al describe the role of TBX3 in BRAF^{V600E}-driven papillary thyroid cancer progression. They find that the BRAF/MAPK pathway activation induces TBX3 expression through AP-1 mediated transcriptional regulation. Moreover, they identify CXCR2 ligands Cxcl1, 2 and 8 expression to be increased by TBX3 and mediating tumor cell growth.

The identified mechanism is of great relevance for the understanding of BRAF^{V600E}-driven PTC and the experiments in the different mouse and human model systems are very well designed. The results throughout are convincing. However, the manuscript should be revised for English language. While the data is convincing, it would help if the figures would be annotated in more detail to be more self-explanatory. A strength of the paper is the consistent results in different models of thyroid cancer. However, the genotype of the human cell lines (K1, TPC1 and BCPAP) should be stated.

Response: Thank the reviewer for the suggestion. We have improved our writing and corrected the figures according to your suggestion. The key genetic drivers of K1 and BCPAP are BRAF^{V600E}, however TPC1 harbors a CCDC6-RET fusion which was generated by intrachromosomal inversion¹³. Both BRAF^{V600E} and CCDC6-RET fusion results in MAPK pathway activation, which was described in our manuscript.

Manuscript revision was made on page 7 line 158 to 160: “Similar as observed in mouse models, over-expression of wild-type BRAF or BRAF^{V600E} in normal thyroid and PTC cells (K1 [BRAF^{V600E}] and TPC1 [BRAF^{WT}]) resulted in up-regulation of TBX3 (Fig. 2a).”

Specific Comments:

1.) Abstract and Title:

- The authors use the term multikinase inhibitors too loosely, which can be misleading. In their manuscript they study the effect of targeted BRAF-inhibition with vemurafenib, which is relatively selective for RAF kinases. The term multikinase inhibitors is better applied for compounds that target multiple kinases (i.e. VEGFR and others).

Response: Thank the reviewer very much for the suggestion. As the reviewer pointed out, multiple kinase inhibitors (MKIs) represent a broad range of drugs, which include Multitargeted Tyrosine Kinase Inhibitors (MTKIs) and MAPK inhibitors (MAPKis)^{14,15}. In the current study, we inhibited MAPK signaling pathway with MAPK inhibitor, mainly BRAF inhibitor PLX4032, and ERK

inhibitor SCH772984, so it will be more accurate to use “MAPKi” here. We have made corresponding corrections.

Manuscript revision was made on the title and abstract. We replaced MKIs with MAPKi.

2.) Introduction

- Page 3, line 45-47: As above. The authors should distinguish between multikinase inhibitors such as Sorafenib and Lenvatinib, which have shown activity against radioiodine refractory thyroid cancer (Brose MS et al, Lancet 2014; Schlumberger M et al, NEJM, 2015) and the combined BRAF/MEK inhibition in BRAF^{V600E} mutant PTCs, which has more modest activity and has not been approved by the FDA for this indication.

Response: We agree with the reviewer that comprehensive understanding of clinical situation for thyroid cancer is very important for researchers to generate significant scientific questions. As the reviewer has mentioned, MKIs including Lenvatinib and Sorafenib have been approved by FDA for the treatment of metastatic, RAI-refractory DTC with ~65% CR+PR (CR, complete responses; PR, partial response) and ~12% PR, which trial finally discontinued due to adverse effects^{16,17}.

As MAPK inhibitors are concerned, FDA approved BRAF/MEK inhibitors Dabrafenib and Trametinib for the treatment of ATC in May 2018¹⁸. PLX4032, or Vemurafenib, as a selective inhibitor of BRAF^{V600E}, has been FDA approved for treatment of BRAF^{V600E}-mutant melanoma. However, the clinical benefit of Vemurafenib/Dabrafenib on BRAF^{V600E}-positive RAI-refractory or metastatic DTC patients was not acceptable compared to the toxicity profile. Other therapeutic strategies based on combination of BRAF inhibitors and other targeted agents are also under pre-clinical or clinical development in thyroid carcinoma. The MEK1/2 inhibitor AZD6244 sensitizes BRAF^{V600E} thyroid cancer to Vemurafenib, but the study is restrained at the preclinical level¹⁹. Multiple studies have shown that the transient response followed with refractoriness to BRAF/MEK inhibition could be due to activation of alternative pathways. Thus, more combined therapies, even with immune-therapies are still on the way. We really hope we could provide more basic support for this field in the future.

Manuscript revision was made on page 3 line 45-48: “Nevertheless, targeted therapies with MAPK inhibitor drugs (MAPKi), including Vemurafenib

or Dabrafenib, exhibited reserved efficacy on BRAF^{V600E}-positive metastatic, RAI-refractory DTC patients due to primary or acquired drug resistance.”

3.) Results

Figure 1:

- Heading: Stating that Tbx3 is necessary for Braf^{V600E} to induce PTC seems an overstatement based on the data presented.

Response: Thank the reviewer for the kind suggestion. We have changed the heading to “Loss of Tbx3 inhibited Braf^{V600E}-induced PTC initiation and progression.”

Related revision was made on the heading for Fig. 1 and Result 1.

- Figure 1f and g: the results are difficult to read. The interconnecting lines between genotypes are not explained. Is this intended to show difference between littermates? It would help, if the authors would represent the thyroid weights and mouse weights similar in figure 1k, or in boxplots.

Response : Thank the reviewer for the question. To better demonstrate that the dose of TBX3 has an important effect on tumor size and body weight, we conducted the comparison among litter-mates and each line represents one litter. To appreciate the data more clearly, we switched the graphs to scatter plot and updated Fig. 1d and 1f

Related revision was made on Fig. 1d and 1f. We performed the tumor weight and body weight comparison across three genotypes and summarized in Fig.1d and 1f.

- In the IHC images in 1e and 1j the differences between the different mouse models are challenging to identify. Would the authors be able to either provide better quality pictures (perhaps with an inset at higher magnification) and/or quantify the Ki67-staining?

Response : Thank the reviewer very much for the suggestion. We are pretty sure that deletion of Tbx3 repressed tumor progression and reduced Ki67 level within mPTC and mPTC-TAM models. For better presentation, we have provided higher quality pictures in Fig. 1e and 1j. The Ki67 quantification results were included as Supplementary Fig.1f and 1l.

Related revision was made on Fig. 1e, 1j and Supplementary Fig. 1f, 1l.

We have provided higher quality pictures in Fig. 1e and 1j. The Ki67 quantification results were included as Supplementary Fig. 1f and 1l.

Figure 2:

- Figure 2a: Could the authors provide the rationale for overexpressing BRAF in the BRAF^{V600E} mutant K1 cell line? This line is more appropriate for the KD experiment shown in 1b. Moreover, the genetic driver of the TPC1 cell line is known to be a RET fusion, meaning that MAPK pathway is constitutively activated but subject to ERK feedback. What is the rationale for creating a BRAF overexpressing TPC1 cell line?

Response : We agree with the reviewer that most PTC cell lines, including K1, BCPAP and KTC-1, harbor BRAF^{V600E} mutation, and TPC1 carries a CCDC6-RET fusion. The MAPK pathway should be constitutively activated in both K1 and TPC1 cells. As we checked the expression of TBX3 through a batch of PTC cell lines, we noticed the TBX3 level in K1 is higher than that in TPC1 cells. Besides, the TBX3 level in Nthy is hardly detectable, where BRAF/MAPK activity is relatively normal¹. We hypothesize that distinct TBX3 levels could relate to different degrees of BRAF/MAPK activation. Indeed, the basal p-ERK1/2 level in K1 is higher than that in TPC1 or Nthy cells. Therefore, we simply over-expressed BRAF/BRAF^{V600E} to see what over-dosed BRAF/BRAF^{V600E} will do, in terms of TBX3 expression. Clearly, the influence of BRAF/BRAF^{V600E} over-expression is different, with subtle up-regulation of TBX3 in K1 and TPC1, but significant up-regulation in Nthy cells, which builds the strong correlation between BRAF/MAPK activation and TBX3 expression.

While addressing this question, we are thinking the better design could be over-expressing BRAF/BRAF^{V600E} on BRAF-KD cells (K1 or TPC1) to see the rescue effect. We will definitely follow this up in the future since it is also important for further mechanism understanding.

- Figure 2l-m) please add for clarification the model system/cell line to the respective graph.

Response : Thank the reviewer very much for your suggestion. We have made corrections to Fig. 2l-n.

Related revision was made on Fig. 2l-n: We marked the model system to the graphs.

Figure 3:

- Figure 3b: Please add a title to the graphs showing what is compared to what. Please provide the list of the 191 genes overlapping between mouse and human.

Response: Thank the reviewer for the advice. Fig. 3b showed the venn diagram analysis between two groups of DEGs from mPTC tissues versus K1 cells upon TBX3 loss. The title “Co-regulated genes by TBX3 in different species” was added to Fig. 3b.

The list of the 191 co-regulated genes by TBX3 in mouse and human was showed in Supplementary Table 1 cited on page 9 line 207.

Related revision was made on Fig. 3b. The title “Co-regulated genes by TBX3 in different species” was added to Fig. 3b.

- Figure 3e: consider breaking up the Y-axis to better display the differences between shCtrl and the two TBX3 short hairpins for all genes. The increase in *Pdgfb* is intriguing. Was there any difference in endothelial or stromal cell elements in the *Tbx3*-null tumors?

Response :Thank the reviewer for pointing out the question. We have updated Fig. 3e according to your advice.

Previous study has demonstrated that PDGF-BB ligand was found in the majority of thyroid tumors²⁰. Our preliminary *in vitro* data has shown some hint of possible involvement of TBX3 in tumor angiogenesis, but we agree with the reviewer, it will be necessary to check endothelial or stromal composition in *Tbx3*-null tumors. Based on the critical function of *Tbx3* (as well as *Tbx2*) in the embryonic outflow duct formation^{21,22}, as well as glomerulus blood vessel formation²³, it will be worthwhile investigating the significance of TBX3 in angiogenesis through various tumorigenesis contexts with specific genetic or pharmacological tools.

Related revision was made on Fig. 3e. We broke up the Y-axis of Fig. 3e.

- Figure 3f: The differences between shctrl and shTBX3 seem to be very subtle and they are not well shown in the image provided.

Response: Thank the reviewer for pointing out the question. We have replaced Fig. 3f with a more representative picture, and the original data was

shown in Supplementary Table 2.

Related revision was made on Fig. 3f.

Figure 4:

- Figure 4a and b: both figures are very hard to interpret, since it is not clearly visible what is compared to what.

Response : Thank the reviewer for kindly pointing out the question. We found the position of “Vector” and “TBX3” need to be moved left which was renewed in Fig 4a. We here showed CXCR2 ligands RNA level of “DMSO+TBX3-overexpressing” compared to “DMSO+Vector”, then we compared “inhibitor-treated TBX3-overexpressing” to “DMSO+TBX3-overexpressing”. To appreciate the data better, we show the result with three graphs and move the data for CXCL2 and CXCL8 into Supplementary Fig.4a. Similar revision was made for Fig. 4b.

Related revision was made on Fig. 4a, 4b and Supplementary Fig. 4a and 4d.

- Figure 4d: Why is breast cancer and thyroid cancer compared at this point? Please provide the list of the overlapping 76 genes.

Response : To find out the molecular mechanism by which TBX3 activates IKK β /NF- κ B pathway, we analyzed potential genes that TBX3 might regulate in the database. Current available TBX3 target-dataset was only found in a research about breast cancer, where they analyzed up-regulated genes upon TBX3 over-expression. In fact, we found that TBX3 over-expression in breast cancer also up-regulated some CXCR2 ligands, suggesting that there could be a similar regulatory mechanism between both types of tumors. Therefore, we converged the breast cancer data and our PTC RNA-seq data aiming to find co-regulated genes. The list of the overlapping 76 genes was shown in Supplementary Table 3.

Figure 6: The authors describe a reduction in ‘G-MDSC’ in the context of TBX3^{-/-} mPTCs. However, the authors define MDSCs in general as cells expressing CD45⁺CD11b⁺Gr1⁺. Cells expressing the aforementioned markers can be a wide variety of cells in the tumor microenvironment such as Monocytes, Neutrophils, G-MDSC, M-MDSC, immature Macrophages etc. While it is interesting that the authors find a difference in abundance in the

context of TBX3^{-/-}, based on their data the assumption that this population represents MDSCs cannot be made without additional characterization. Since the authors have performed bulk RNA seq on the whole mouse tissue, they could perform immune deconvolution to identify the presence of different immune populations.

Response : Thank the reviewer for your question. In mice, MDSCs historically were defined as cells expressing both Gr-1 and CD11b markers, and two subpopulations are characterized: G-MDSC(CD11b⁺Ly6G⁺Ly6C^{lo}) and M-MDSC (CD11b⁺Ly6G⁻Ly6C^{hi}) which are widely accepted and used to analyze MDSC in mice ^{11,24}. When we performed the FACS array on different mPTC tumor tissues, we used CD45⁺CD11b⁺Gr1⁺ to mark total MDSC population, but distinguished G-MDSC from M-MDSC by using different marker combinations (CD11b⁺Ly6G⁺Ly6C^{lo} for G-MDSC; CD11b⁺Ly6G⁻Ly6C^{hi} for M-MDSC) in the following assays (Fig. 6b and 6e). In addition, we did MDSCs-T cell suppression and G-MDSCs chemotaxis experiments during the revision, which showed that G-MDSCs are attracted into TME and suppress T cell function (Supplementary Fig.7a and 7f).

According to the reviewer's suggestion, we performed immune deconvolution through ImmuneCellAI and found the abundance of neutrophils were reduced in mPTC/Tbx3^{-/-} tumors (**Response Figure 1**), which supported the rational that MDSCs represent pathologically activated neutrophils²⁵.

Response Figure 1 Heatmap of Granulocytes and Neutrophils according to the RNA-seq of whole mouse tissue, mPTC: C_1, C_2, C_3; mPTC/Tbx3^{-/-}: CKO_1, CKO_2, CKO_3.

Figure 8: Zhong et al show rRNA and protein expression of sorted tumor cells from mouse PTC w/o TBX3^{-/-}. These are important results, that should be included earlier in figure 1 or 2. Moreover, since the hypothesis is that the mechanism by which TBX3^{-/-} mediates progression is cell autonomous, would

it be possible to compare RNA seq from sorted tumor cells with human K1 cells instead of whole mouse tissue with K1 cell line as performed in figure 3b?

Response : Thank the reviewer very much for your advice. We moved Fig. 8a, 8b and Supplementary Fig. 8a, 8b to Supplementary Fig. 3c-f. We analyzed the molecular changes caused by TBX3 knockout in primary mPTC cells obtained from mPTC^{GFP} model. The results showed that TBX3 knockout significantly inhibited the expression of CXCR2 ligands, which was consistent with our results from the mPTC tumor tissue. The finding was assigned as Supplementary Fig. 3g.

We also tried to do RNA-seq on sorted tumor cells, but we had a hard time to get enough qualified RNA samples due to small cell numbers got from limited mice tissues. Actually, we sent as many RNA samples as we can for library construction, but only a small number of libraries were achieved. For the future, we will continue this part of work, and hope we will get enough tumor cells through extending the tumor growth cycle.

The recent paper published on Nature Communication by Wang *et al* illustrated scRNA-seq of PTC patient samples, also indicated considerable infiltration of myeloid cells in patient tumors²⁶, which conclusion is analogous to ours. Further understanding about advanced PTC TME composition will benefit from enlarged sample size of analysis.

Supplementary figures:

Suppl. Figure 1:

- 1a) What about the DEGs in mouse and human PTCs or cell lines, which would be more relevant to this study than gene expression changes in melanomas? See Montero-Conde C. Cancer Discov 2013 for human thyroid cancer cell lines; Dunn L. JCEM 2018 for patient biopsy specimens; Saqcena M Cancer Discov 2021, for GEMM of Braf PTCs.

Response : We agree with the reviewer that comparison of DEGs in mouse and human PTCs or cell lines will be very relevant to the current context. As mentioned in multiple reviews and Montero-Conde C's paper, melanoma cells normally respond well to MAPKi, while thyroid cancer cells are relatively refractory due to activation of alternative pathways. So we took advantage of datasets from MAPKi treated melanoma patients and cell lines, where we found the significant involvement of TBX3. We actually went back to the DEGs in mPTC and PTC cell lines treated with MEK or BRAF^{V600E} inhibitor from Montero-Conde C's paper (GSE37441), and Saqcena M's paper

(GSE147479), where TBX3 was not always repressed. Similarly, a batch of well-known MAPK signaling responsive factors, such as c-Jun, Junb, c-Fos, Fosb, Elk1, Myc, Stat3 were not always down-regulated in MAPKi treated cells or mouse tissues either. But our following *in vivo* and *in vitro* studies consistently confirm the important function of TBX3 in PTC advancement. Thus the melanoma datasets could provide valuable information in terms of BRAF/MAPK downstream factors.

For the future, it will be necessary to set up an optimal system for identifying instant and direct responsive genes to BRAF/MAPK, and also for finding indispensable genes in PTC initiation and progression. Hopefully, specific targeting of passenger genes, such as TBX3, downstream of critical driver events like BRAF/MAPK signaling activation will provide more customized therapeutic strategies for different individuals.

- 1e) In the y axis: relative Tbx3 mRNA

Response : Thank the reviewer for pointing out inaccurate description, we have corrected the y axis in Supplementary Fig. 1e.

Related revision was made on Supplementary Fig. 1e.

- 1e-g): Please add comparison to WT thyroid

Response : Thank the reviewer for your suggestion, we have showed the comparison between WT and mPTC at Fig. 1b and Supplementary Fig. 1c, so here we only showed the difference between mPTC and mPTC/Tbx3^{-/-}. To avoid data duplication, we did not include the comparison with WT here.

Reviewer #4 (Remarks to the Author): with expertise in MDSC, cancer

Manuscript “Targeting a $\text{Braf}^{\text{V600E}}$ -TBX3-MDSC Axis Reverts Immune Suppression and Sensitizes Thyroid Carcinoma to MKIs Therapy” by Zhang et al, aims to investigate: 1) the role of developmental factor TBX3 in mediating the oncogenic functions of $\text{Braf}^{\text{V600E}}$ mutation in thyroid cancer and 2) the immunoregulatory effects of TBX3-directed TLR2-NFKB-CXCR2 ligand signaling on the recruitment of MDSCs to tumors. The manuscript utilizes multiple in vivo models, clinical data and in vitro platforms to substantiate the primary role of TBX3, and its down-stream targets TLR2-NFKB-CXCR2, in the initiation and progression of $\text{Braf}^{\text{V600E}}$ -induced papillary thyroid cancer (PTC) and in the infiltration of MDSCs to PTC tumors. The induction of TBX3 in thyroid cancer cells was the result of the activation of the $\text{Braf}^{\text{V600E}}$ -MAPK-AP1 axis. Overall, this manuscript is timely and provides significant insight into the previously underexplored role of TBX3 in thyroid cancer and contributes to the field’s understanding of immunoregulatory axes in $\text{Braf}^{\text{V600E}}$ tumors. Also, results could serve as the foundation for new therapies for PTC. Noticed limitations are as follows:

Major comments/suggestions:

1) Discrimination of the effects of tumor cell related CXCR2 vs. stroma-myeloid cells expressed CXCR2 in the growth of $\text{Braf}^{\text{V600E}}$ -induced PTC in mice remains unknown. Although investigators show results demonstrating the crucial role of tumor cell related CXCR2 in the growth of PTC tumors, evaluating the interaction of tumor produced CXCL1, CXCL2 and CXCL8 with the CXCR2 in the stroma would be highly complementary.

Response: Thank the reviewer very much for the suggestion. Similar as other reports²⁸, our IF staining showed that CXCR2 is more abundantly expressed on MDSCs than tumor cells (Supplementary Fig. 7d).

As the tumor cell CXCR2 is concerned, it mediates the autocrine pro-proliferation function of ligands (CXCL1, 2, 8) since inhibitor SB265610 effectively inhibited the proliferation of PTC tumor cells *in vitro*. This is consistent with previous reports where CXCR2 ligands promote tumor cell proliferation, maintain stemness, and enforce EMT in an autocrine manner²⁹. As shown in bellow, we found that *Tbx3* loss mainly repressed CXCR2 ligands,

but hardly affect CXCR2 expression in tumor cells through RT-PCR (**Response Figure 2**). Therefore, decreased autocrine-dependent tumor cell proliferation caused by CXCR2 ligands reduction explains one part of the phenotype in mPTC/Tbx3^{-/-}.

On the other hand, the CXCR2 on MDSCs surface responds to ligands secreted by tumor cells. To prove this, we performed chemotaxis experiment by co-culturing mPTC primary tumor cells with G-MDSCs from spleen and bone marrow, and we found CXCR2 inhibitor SB265610 blocks migration of G-MDSCs. Moreover, reduced CXCR2 ligands (mainly Cxcl1 and Cxcl2 in mouse cells) caused by Tbx3 loss in mPTC/Tbx3^{-/-} primary tumor cells inhibited chemotaxis of G-MDSCs, which was further blocked by SB265610 (Supplementary Fig. 7e). Thus, defected recruitment of MDSCs into the TME caused by Tbx3 loss associated CXCR2 ligands reduction explains another part of the phenotype in mPTC/Tbx3^{-/-}.

Together, in our *in vivo* SB265610 treatment experiment, reduced both tumor cell proliferation and MDSCs abundance were observed, which led to inhibited tumor growth in Fig. 6h-j. We conclude that TBX3-CXCR2 ligands axis promotes tumor cell proliferation by autocrine, and immunosuppression by paracrine recruitment of MDSCs.

Response Figure 2 a. CXCR2 expression level in Normal (Nthy) and PTC cells (TPC1 and K1). **b.** CXCR2 level in K1 cells with TBX3 knockdown.

Related revision was made on Supplementary Fig. 7e and legends IF staining of CXCR2 on mPTC tumor tissues was included in Supplementary Fig. 7e.

Manuscript revision was made on page 13 line 321 to 325: “As expected, conditioned media of primary mPTC tumor cells attracted MDSCs in chemotaxis assay, which was dramatically diminished by Tbx3 removal

(Supplementary Fig. 7e). While chemical interruption of CXCR2-ligands communication by SB265610 repressed tumor cell-induced MDSCs migration significantly (Supplementary Fig. 7e).”

2) It remains unclear how the upregulation of CXCR2 in Braf^{V600E}-TBX3-induced papillary thyroid tumors promoted the infiltration of PMN-MDSCs. Is CXCR2 signaling in tumor cells promoting the expression of factors that recruit PMN-MDSCs? Is the TBX3 driven production of CXCL1, CXCL2 and CXCL8 attracting CXCR2+ PMN-MDSCs? This should be tested experimentally.

Response: Thank the reviewer very much for the question. We did not detect upregulation of CXCR2 during Braf^{V600E}-TBX3-induced papillary thyroid tumorigenesis, instead, we observed increased CXCR2 ligands such as CXCL1, CXCL2 and CXCL8 (Supplementary Fig. 9b-d). So we propose that BRAF/MAPK pathway activation promotes the infiltration of PMN-MDSCs through increasing Tbx3-CXCR2 ligands (mainly CXCL1, 2, 8) cascade, where the ligands attract CXCR2⁺ PMN-MDSCs into the PTC TME.

On one hand, BRAF^{V600E} induced BRAF/MAPK pathway activation up-regulates TBX3 expression, which in turn increases CXCR2 ligand levels (related to Fig. 8a-b and Supplementary Fig. 9a). Thus, ligand variations caused by TBX3 over-expression or knock-down in tumor cells affect MDSCs recruitment in the chemotaxis experiments.

On the other hand, ligands bind to CXCR2 expressed on MDSCs surfaces to recruit them into the TME. So the inhibitor SB265610 blocks MDSCs attraction mediated by PTC cell-derived conditioned medium.

Related experiments regarding the involvement of BRAF/MAPK/Tbx3-regulated CXCR2 ligands on MDSCs attraction by tumor cells, as well as the critical necessity of CXCR2 were included as Supplementary Fig. 7e-7g.

Related revision was made on Supplementary Fig7e-7g and legends.

Manuscript revision was made on page 13 line 321-328: “As expected, conditioned media of primary mPTC tumor cells attracted MDSCs in chemotaxis assay, which was dramatically diminished by Tbx3 removal (Supplementary Fig. 7e). While chemical interruption of CXCR2-ligands communication by SB265610 repressed tumor cell-induced MDSCs migration

significantly (Supplementary Fig. 7e). Similarly, conditioned medium from TBX3 knock-down PTC cells showed impaired neutrophil attraction, which was rescued by over-expressed CXCR2 ligands (Supplementary Fig. 7f). SB265610 treatment significantly suppressed recruitment of neutrophils as well (Supplementary Fig. 7g).”

3) Authors refer to MDSCs without developing functional assays of T cell suppression. It is important to validate the suppressive function of MDSCs and determine if TBX3 deletion modulates only the accumulation or also the direct T cell suppressive potential of PMN-MDSCs.

Response: Thank the reviewer for the question. *In vitro*, co-culture experiments showed that primary MDSCs from mPTC are capable of suppressing CD8⁺ T cell proliferation. By comparison, we didn't detect significance compromise of the suppressive potential in mPTC/TBX3^{-/-} MDSCs. Considering the reduced infiltration ratio of mPTC/TBX3^{-/-} MDSCs, we prefer to believe Tbx3 deletion affected more of the accumulation rather the suppressive function of MDSCs.

Related revision was made on Supplementary Fig. 6g and legends. We compared the suppression function of MDSCs from mPTC and mPTC/TBX3^{-/-} tumor tissues

Manuscript revision was made on page 13 line 304-306: “Co-culture assay showed that G-MDSCs from mPTC were capable of suppressing CD8⁺ T cell proliferation, which was hardly compromised by Tbx3 knock-out (Supplementary Fig. 6g).”

4) The impact of thyroid cancer cell TBX3 in the immunosuppressive myelopoiesis occurring systemic remains unclear. Establish whether TBX3 expression impacts development of MDSCs from bone marrow precursors or if its effects are predominantly via recruitment is crucial.

Response: Thank the reviewer very much for the professional question. Based on our quantification, bone marrow MDSCs and granulocyte-monocyte progenitors (GMPs) were not affected by Tbx3 knockout from thyroid cancer

cells. But the circulation and splenic G-MDSCs abundance were decreased in mTPC/Tbx3^{-/-} mice. We agree with the idea that TBX3 expression impacts MDSCs distribution predominantly via recruitment. For this mechanism, we have three supporting evidences. First, we detected decreased CXCL1 and CXCL2 levels in the mTPC/Tbx3^{-/-} serum. Second, conditioned media of primary mPTC tumor cells attracted G-MDSCs from both bone marrow and spleen in chemotaxis assay, which was dramatically diminished by Tbx3 removal due to CXCL1 and CXCL2 reduction. Third, the migration of MDSCs toward conditioned media is CXCR2 dependent since CXCR2 inhibitor SB265610 blocked the migration. Together, Tbx3-CXCR2 ligands axis mediates the recruitment of MDSCs into mPTC rather than affecting the myelopoiesis.

Related revision was made on Supplementary Fig. 7a-c and legends The contents of G-MDSCs and M-MDSCs in bone marrow as well as peripheral blood and spleen from mPTC and mTPC/Tbx3^{-/-} mice were compared and presented in Supplementary Fig. 7a. Additionally, the bone marrow precursors for MDSCs were compared and showed in Supplementary Fig. 7b. Levels of CXCL1 and CXCL2 in blood and supernatants of primary tumor cells were measured by ELISA in Supplementary Fig. 7c.

Manuscript revision was made on page 13 line 314 to 320: “Interestingly, circulation and splenic G-MDSCs abundance were also decreased in mTPC/Tbx3^{-/-}, while bone marrow MDSCs and granulocyte-monocyte progenitors (GMPs) remain constant (Supplementary Fig. 7a, b). Considering the intermediary function of ligands-CXCR2 interaction between cell communications, we wondered whether compromised MDSCs infiltration in mutants is due to CXCR2 ligands reduction caused attraction deficit. Indeed, the secretion of CXCL1 and CXCL2 was significantly suppressed in mTPC/Tbx3^{-/-}, concurrent with reduced serum CXCL1 and CXCL2 levels (Supplementary Fig. 7c).”

5) Elucidating whether elimination of T cells (or development of experiments in T cell deficient mice) overcomes the anti-tumor effect induced by TBX3 deletion in Braf^{V600E}-induced PTC will confirm the key role of T cells in the

reported anti-tumor responses and the interaction of tumor cell TBX3 and T cell dysfunction. This should be complemented with expression of cytotoxic mediators on T cells.

Response: Thank the reviewer very much for your suggestion. To further determine the important role of CD8⁺ T cells in mPTC progression, we suppressed CD8⁺ T in mPTC using CD8 antibody. Anti-CD8 rescued the tumor formation in mPTC/Tbx3^{-/-} mice efficiently, suggesting that the toxic effect of CD8⁺ T cells may be the main reason for the reduction of mPTC/Tbx3^{-/-} tumors. Interestingly, anti-CD8 did not significantly affect the tumor volume in mPTC mice, probably due to the highly suppressed T cell activity caused by enriched MDSCs.

Related revision was made on Fig. 6g, Supplementary Fig. 6i and legends

The therapeutic effect of anti-CD8 on mPTC growth was presented as Fig. 6g. Expression of cytotoxic mediators on T cells was presented as Supplementary Fig. 6i.

Manuscript revision was made on page 13 line 308-312: “To further valuate the anti-tumor effect of CD8⁺ T cell in Braf^{V600E}-induced mPTC, we depleted CD8⁺ T cell with anti-CD8 antibody. Remarkably, the anti-tumor effects of CD8⁺ T cells were abrogated specifically in mPTC/Tbx3^{-/-} with recovered tumor growth, since cytotoxic CD8⁺ T cell were significantly depleted compared to mPTC (Fig. 6g and Supplementary Fig. 6i).”

6) It remains unknown whether the higher numbers of T cells present in TBX3-null tumors carry higher expression of checkpoint mediators. This could open the possibility of combination strategies with checkpoint inhibitors.

Response: Thank the reviewer very much for your suggestion. No significant difference was found for TCRβ⁺CD8⁺PD1⁺ and TCRβ⁺CD8⁺PD1⁺TIM-3⁺ cells between mPTC and mPTC/Tbx3^{-/-} tumors (**Response Figure 3**).

Response Figure 3 Percentage of TCR β^+ CD8⁺PD1⁺ T and TCR β^+ CD8⁺TIM3⁺ T cells of total CD45⁺ TILs in mPTC and mPTC/Tbx3^{-/-} tumors was analyzed by FlowJo, $n=4$.

Minor comments/suggestions:

1) Densitometric quantification of western blots is suggested.

Response: Thank the reviewer for the suggestion. We marked the molecular weights of target bands into each western blot panel. We also performed densitometric analysis and presented all related panels in Supplementary Fig. 10-12.

2) Clarify differences in tumor growth in Figure 5f for CXCL1/2 and CXCL8 overexpression models- Why does OE of CXCL2 or CXCL1 provoke the same effects while CXCL8 OE does not

>Do CXCL1/2 and CXCL8 exhibit differential affinity for CXCR2?

>Do CXCL1/2 and CXCL8 activate the same downstream effects at the same magnitude?

Response: We also noticed the different function on tumor proliferation between CXCL1/2 and CXCL8 over-expression. We agree with the reviewer that one reason could be differential binding affinity for CXCR2. Similar phenomenon was actually demonstrated in ours and others study, where over-expression of different ligands showed different attraction capabilities on MDSCs (Supplementary Fig. 7f) ^{30,31}. Since the same receptor, which is CXCR2, mediates the interaction with CXCL1/2 and CXCL8, we prefer to believe the same downstream signalings are initiated and transduced.

3) Include the empty vector + CXCL1/2/8 tumor growth curves/weights in the same figures alongside the shTBX3 OE data (Figure 5f)

Response: Thank the reviewer very much for your suggestion. We have re-organized the figure.

Related revision was made on Fig. 5e.

References

- 1 Li, X. M. et al. TBX3 promotes proliferation of papillary thyroid carcinoma cells through facilitating PRC2-mediated p57(KIP2) repression. *Oncogene* **37**, 2773-2792 (2018).
- 2 Charles, R. P., Iezza, G., Amendola, E., Dankort, D. & McMahon, M. Mutationally Activated BRAF(V600E) Elicits Papillary Thyroid Cancer in the Adult Mouse. *Cancer Research* **71**, 3863-3871 (2011).
- 3 Byun, E., Park, B., Lim, J. W. & Kim, H. Activation of NF-kappaB and AP-1 Mediates Hyperproliferation by Inducing beta-Catenin and c-Myc in Helicobacter pylori-Infected Gastric Epithelial Cells. *Yonsei Med J* **57**, 647-651 (2016).
- 4 Sun, Y. et al. Inflammatory signals from photoreceptor modulate pathological retinal angiogenesis via c-Fos. *J Exp Med* **214**, 1753-1767 (2017).
- 5 Hu, H. et al. CRL4B catalyzes H2AK119 monoubiquitination and coordinates with PRC2 to promote tumorigenesis. *Cancer Cell* **22**, 781-795 (2012).
- 6 Kalhori, V. & Tornquist, K. MMP2 and MMP9 participate in S1P-induced invasion of follicular ML-1 thyroid cancer cells. *Mol Cell Endocrinol* **404**, 113-122 (2015).
- 7 Liu, H., Deng, H., Zhao, Y., Li, C. & Liang, Y. LncRNA XIST/miR-34a axis modulates the cell proliferation and tumor growth of thyroid cancer through MET-PI3K-AKT signaling. *J Exp Clin Cancer Res* **37**, 279 (2018).

- 8 He, Q. et al. SIX4 promotes hepatocellular carcinoma metastasis through upregulating YAP1 and c-MET. *Oncogene* **39**, 7279-7295 (2020).
- 9 Bagheri-Yarmand, R. et al. Combinations of Tyrosine Kinase Inhibitor and ERAD Inhibitor Promote Oxidative Stress-Induced Apoptosis through ATF4 and KLF9 in Medullary Thyroid Cancer. *Mol Cancer Res* **17**, 751-760 (2019).
- 10 Li, Y. et al. KLF9 suppresses gastric cancer cell invasion and metastasis through transcriptional inhibition of MMP28. *FASEB J* **33**, 7915-7928 (2019).
- 11 Bronte, V. et al. Recommendations for myeloid-derived suppressor cell nomenclature and characterization standards. *Nat Commun* **7**, 12150 (2016).
- 12 Tanriover, G., Eyinc, M. B., Aliyev, E., Dilmac, S. & Erin, N. Presence of S100A8/Gr1-Positive Myeloid-Derived Suppressor Cells in Primary Tumors and Visceral Organs Invaded by Breast Carcinoma Cells. *Clin Breast Cancer* **18**, e1067-e1076 (2018).
- 13 Ishizaka, Y., Ushijima, T., Sugimura, T. & Nagao, M. cDNA cloning and characterization of ret activated in a human papillary thyroid carcinoma cell line. *Biochem Biophys Res Commun* **168**, 402-408 (1990).
- 14 Naoum, G. E., Morkos, M., Kim, B. & Arafat, W. Novel targeted therapies and immunotherapy for advanced thyroid cancers. *Mol Cancer* **17**, 51 (2018).
- 15 Bikas, A., Vachhani, S., Jensen, K., Vasko, V. & Burman, K. D. Targeted therapies in thyroid cancer: an extensive review of the literature. *Expert Rev Clin Pharmacol* **9**, 1299-1313 (2016).
- 16 Schlumberger, M. et al. Lenvatinib versus placebo in radioiodine-refractory thyroid

- cancer. *N Engl J Med* **372**, 621-630 (2015).
- 17 Brose, M. S. et al. Sorafenib in radioactive iodine-refractory, locally advanced or metastatic differentiated thyroid cancer: a randomised, double-blind, phase 3 trial. *Lancet* **384**, 319-328 (2014).
- 18 Subbiah, V. et al. Dabrafenib and Trametinib Treatment in Patients With Locally Advanced or Metastatic BRAF V600-Mutant Anaplastic Thyroid Cancer. *J Clin Oncol* **36**, 7-13 (2018).
- 19 Song, H. et al. The MEK1/2 Inhibitor AZD6244 Sensitizes BRAF-Mutant Thyroid Cancer to Vemurafenib. *Med Sci Monit* **24**, 3002-3010 (2018).
- 20 Adewuyi, E. E. et al. Autocrine activation of platelet-derived growth factor receptor alpha in metastatic papillary thyroid cancer. *Hum Pathol* **75**, 146-153 (2018).
- 21 Xie, H. L. et al. Identification of TBX2 and TBX3 variants in patients with conotruncal heart defects by target sequencing. *Hum Genomics* **12** (2018).
- 22 Mesbah, K., Harrelson, Z., Theveniau-Ruissy, M., Papaioannou, V. E. & Kelly, R. G. Tbx3 is required for outflow tract development. *Circ Res* **103**, 743-750 (2008).
- 23 Barry, D. M. et al. Molecular determinants of nephron vascular specialization in the kidney. *Nature Communications* **10** (2019).
- 24 Zhou, J., Nefedova, Y., Lei, A. H. & Gabrilovich, D. Neutrophils and PMN-MDSC: Their biological role and interaction with stromal cells. *Semin Immunol* **35**, 19-28 (2018).
- 25 Miao, Y. R. et al. ImmuCellAI: A Unique Method for Comprehensive T-Cell Subsets Abundance Prediction and its Application in Cancer Immunotherapy. *Adv Sci (Weinh)*

- 7, 1902880 (2020).
- 26 Pu, W. et al. Single-cell transcriptomic analysis of the tumor ecosystems underlying initiation and progression of papillary thyroid carcinoma. *Nat Commun* **12**, 6058 (2021).
- 27 Saqcena, M. et al. SWI/SNF complex mutations promote thyroid tumor progression and insensitivity to redifferentiation therapies. *Cancer Discov* (2020).
- 28 Cheng, Y., Ma, X. L., Wei, Y. Q. & Wei, X. W. Potential roles and targeted therapy of the CXCLs/CXCR2 axis in cancer and inflammatory diseases. *Biochim Biophys Acta Rev Cancer* **1871**, 289-312 (2019).
- 29 Nagarsheth, N., Wicha, M. S. & Zou, W. P. Chemokines in the cancer microenvironment and their relevance in cancer immunotherapy. *Nature Reviews Immunology* **17**, 559-572 (2017).
- 30 Liao, W. et al. KRAS-IRF2 Axis Drives Immune Suppression and Immune Therapy Resistance in Colorectal Cancer. *Cancer Cell* **35**, 559-572 e557 (2019).
- 31 Taki, M. et al. Snail promotes ovarian cancer progression by recruiting myeloid-derived suppressor cells via CXCR2 ligand upregulation. *Nat Commun* **9**, 1685 (2018).

Reviewers' Comments:

Reviewer #1:

Remarks to the Author:

The authors have satisfactorily addressed most of my comments but there are some minor comments that they should address:

1. Why have the authors not shown the densitometric readings for the western blots in the main manuscript?
2. The revised sentence in response to comment #4 for the Introduction: "Interestingly, TBX3 was predominantly involved in BRAFV600E-induced MAPK pathway activation and melanoma invasion, which proposes the potential regulation of BRAFV600E on TBX3 through other types of tumorigenesis²⁶." is not very clear and should be rewritten.
3. Under the results section, information on the BRAF and ERK1/2 inhibitors have not been added to the Figure legend 2o.
4. In the figure legend for supplementary figure 1a the information provided for "H" is still not clear. Does H not refer to the third gene set GSE161430?

Reviewer #2:

Remarks to the Author:

The revised manuscript from Zhang et al is greatly improved. I only have minor comments:

- 1.) While the writing is improved there are still inaccuracies throughout the manuscript that should be addressed, e.g:
Line 25: therapies for advanced thyroid cancers remain reserved limited
Line 29-30: by increasing myeloid-derived suppressor cells (MDSCs) penetrance abundance
Line 32, 33 and 35 and multiple other times in the manuscript: MDSC instead of MDSCs
Line 106: a large body of mechanism mechanistic and translational investigations have been conducted
Line 141: 'slightly but significantly reduced': that is contradictory as a statement
Line 142: 'These results showed that Tbx3 dosage is a critical determinant for BRAFV600E-induced thyroid cancer initiation and progression.' I would suggest rephrasing it and include dose-dependent or dose-dependency.
Line 145: from 1month for 1 month
2.) Fig 3f: thank you for updating the figure with a more representative picture. I still find the differences very subtle. Maybe include boxes also in the shCtrl?
- 3.) Fig 4d: I think you are making an important point that the induction of Cxcl2 ligands by TBX3 is not limited to thyroid cancer but also found in breast cancer. I would suggest to mention this in the manuscript.

Reviewer #4:

Remarks to the Author:

Comments have been correctly addressed. There are no additional questions or concerns.

Re: Manuscript NCOMMS-21-20886A

We have undertaken revisions in response to the reviewers' comments and would now like to submit a revised manuscript. The detailed responses and revisions are given below:

Reviewer #1 (Remarks to the Author): with expertise in Tbx3 biology

The authors have satisfactorily addressed most of my comments but there are some minor comments that they should address:

1. Why have the authors not shown the densitometric readings for the western blots in the main manuscript?

Response: Thank the reviewer for the suggestion. We have added all densitometric readings to the major western blot panels of related figures.

2. The revised sentence in response to comment #4 for the Introduction: "Interestingly, TBX3 was predominantly involved in BRAF^{V600E}-induced MAPK pathway activation and melanoma invasion, which proposes the potential regulation of BRAFV600E on TBX3 through other types of tumorigenesis²⁶." is not very clear and should be rewritten.

Response: Thank the reviewer very much for pointing out the unclear statements. We have rewritten the sentences.

Manuscript revision was made on page 4 line 82-87: "Besides, TBX3 was up-regulated by BRAF^{V600E}-induced MAPK pathway activation and promotes melanoma migration via repressing E-cadherin, which correlates TBX3 with BRAF^{V600E} associated tumorigenesis²⁶⁻²⁸. Based on these evidences, to find out the pathological events TBX3 participates in BRAF^{V600E}-induced thyroid tumorigenesis will not only provide us a better understanding of this specific factor, but also clarify the underlying correlation between organ development and tumorigenesis."

3. Under the results section, information on the BRAF and ERK1/2 inhibitors have not been added to the Figure legend 2o.

Response: As you pointed out, we have added information of the BRAF and ERK1/2 inhibitors to the Figure legend 2k.

4. In the figure legend for supplementary figure 1a the information provided for “H” is still not clear. Does H not refer to the third gene set GSE161430?

Response: Thank the reviewer for bringing up this question, and we apologize for the unclear statement. The dataset GSE75299 actually includes transcriptomes upon MAPKi treatment from both melanoma patients and cell lines. Here we used diagram P to represent “down-regulated DEGs analyzed from the patient data part from GSE75299”. We overlapped “down-regulated DEGs analyzed from the cell line data part from GSE75299” and “down-regulated DEGs analyzed from GSE152699” and presented in diagram H.

Related revision was made on figure legend for supplementary figure 1a:

“(a) Venn diagram analysis across three groups of genes. M: Down-regulated DEGs in mouse melanoma cells treated with MAPKi (GSE161430); P: Down-regulated DEGs in melanoma patients treated with MAPKi (GSE75299-patient data part); H: Overlapped down-regulated DEGs from GSE152699 and GSE75299 (cell line data part), referring to human melanoma cell lines treated with MAPKi.”

Reviewer #2 (Remarks to the Author): with expertise in thyroid cancer

The revised manuscript from Zhang et al is greatly improved. I only have minor comments:

1.) While the writing is improved there are still inaccuracies throughout the manuscript that should be addressed, e.g:

Line 25: therapies for advanced thyroid cancers remain reserved limited

Line 29-30: by increasing myeloid-derived suppressor cells (MDSCs) penetrance abundance

Line 32, 33 and 35 and multiple other times in the manuscript: MDSC instead of MDSCs

Line 106: a large body of mechanism mechanistic and translational investigations have been conducted

Line 141: 'slightly but significantly reduced': that is contradictory as a statement

Line 142: 'These results showed that Tbx3 dosage is a critical determinant for BRAFV600E-induced thyroid cancer initiation and progression.' I would suggest rephrasing it and include dose-dependent or dose-dependency.

Line 145: from 1month for 1 month

Response: Thank the reviewer for the suggestions. We corrected the manuscript carefully again. By the way, there seems to be some discrepancy of line numbers, but we tried to find the sentences mentioned by the reviewer and made related revisions.

Manuscript revisions were made on:

Line 22: therapies for advanced thyroid cancers remain reserved

Line 25-27: by increasing myeloid-derived suppressor cells (MDSCs) penetrance

Line 32, 33 and 35 and multiple other times in the manuscript: "MDSC" was replaced with "MDSCs"

Line 103: a large body of mechanism and translational investigations have been conducted

Line 138: slightly but significantly reduced': that is contradictory as a statement

Line 138-140: 'These results showed that Tbx3 determines BRAF^{V600E}-induced thyroid cancer initiation and progression in a dose-dependent way.

Line 142: for 1 month

2.) Fig 3f: thank you for updating the figure with a more representative picture. I still find the differences very subtle. Maybe include boxes also in the shCtrl?

Response: Thank the reviewer for the suggestion. Boxes have been added to the picture representing shCtrl.

3.) Fig 4d: I think you are making an important point that the induction of Cxcl2 ligands by TBX3 is not limited to thyroid cancer but also found in breast cancer. I would suggest to mention this in the manuscript.

Response: Thank the reviewer very much for the suggestion. We have added the related description into the manuscript.

Manuscript revision was made on page 11 line 245 to 248: “While, as a transcriptional activator, TBX3 over-expression elevated CXCR2 ligands in breast cancer as well, indicating similar regulatory mechanisms. We thus compared TBX3-regulated genes in our RNA-seq screening and online breast cancer datasets (Fig. 4d and Supplementary Data 3).”